# Classification of high-dimensional data with spiked covariance matrix structure

**Yin-Jen Chen**                                                              *acgoogwork@gmail.com*
*Department of Statistics*
*North Carolina State University*

**Minh Tang**                                                                  *mtang8@ncsu.edu*
*Department of Statistics*
*North Carolina State University*

**Reviewed on OpenReview:** *https://openreview.net/forum?id=6bQDtTbaQs*

## Abstract

We study the classification problem for high-dimensional data with $n$ observations on $p$ features where the $p \times p$ covariance matrix $\Sigma$ exhibits a spiked eigenvalue structure and the vector $\zeta$, given by the difference between the *whitened* mean vectors, is sparse. We analyze an adaptive classifier (adaptive with respect to the sparsity $s$) that first performs dimension reduction on the feature vectors prior to classification in the dimensionally reduced space, i.e., the classifier whitens the data, then screens the features by keeping only those corresponding to the $s$ largest coordinates of $\zeta$ and finally applies Fisher linear discriminant on the selected features. Leveraging recent results on entrywise matrix perturbation bounds for covariance matrices, we show that the resulting classifier is Bayes optimal whenever $n \to \infty$ and $s\sqrt{n^{-1} \ln p} \to 0$. Notably, our theory also guarantees Bayes optimality for the corresponding quadratic discriminant analysis (QDA). Experimental results on real and synthetic data further indicate that the proposed approach is competitive with state-of-the-art methods while operating on a substantially lower-dimensional representation.

## 1 Introduction

Classification is one of the most important and widely studied inference tasks in statistics and machine learning. Among standard classifiers, the Fisher linear discriminant analysis (LDA) rule is especially popular for its ease of implementation and interpretation. More specifically, suppose that we are given a $p$-variate random vector $\mathbf{Z}$ drawn from a mixture of two multivariate normal distributions $\pi_1 \mathcal{N}_p(\mu_1, \Sigma) + (1 - \pi_1)\mathcal{N}_p(\mu_2, \Sigma)$ and our goal is to classify $\mathbf{Z}$ into one of the two classes. The Fisher LDA rule is then given by

$$\Upsilon_F(\mathbf{Z}) = \begin{cases} 1 & \text{if } \left(\Sigma^{-1}(\mu_2 - \mu_1)\right)^\top \left(\mathbf{Z} - \frac{\mu_1 + \mu_2}{2}\right) \leq \ln \frac{\pi_1}{1 - \pi_1}, \\ 2 & \text{if } \left(\Sigma^{-1}(\mu_2 - \mu_1)\right)^\top \left(\mathbf{Z} - \frac{\mu_1 + \mu_2}{2}\right) > \ln \frac{\pi_1}{1 - \pi_1}, \end{cases} \tag{1.1}$$

provided that $\Sigma$ is positive definite. The Fisher rule $\Upsilon_F$ is the Bayes decision rule, i.e., it achieves the smallest mis-classification error with respect to 0-1 loss for classifying $\mathbf{Z}$. $\Upsilon_F$ is, however, not directly applicable in practice as it involves the unknown parameters $\Sigma^{-1}$, $\mu_1$ and $\mu_2$; we thus usually compute the sample covariance matrix $\hat{\Sigma}$ and the sample means $\bar{X}_1$ and $\bar{X}_2$ from a given training data set with $n$ observations and then plugged these quantities into Eq. (1.1); the resulting classifier is termed as the plug-in LDA rule $\hat{\Upsilon}_F$.

The classification accuracy of $\hat{\Upsilon}_F$ is well-understood in the low-dimensional regime where $p \ll n$. In particular $\hat{\Upsilon}_F$ is asymptotically optimal, i.e., it achieves the Bayes error rate as $n \to \infty$ for fixed $p$. This behavior, however, might no longer holds in the high-dimensional settings $n \asymp p$ or even $n \ll p$ where the sample

covariance matrix $\hat{\Sigma}$ is singular. Indeed, Bickel & Levina (2004) proved that the error rate for $\hat{\Upsilon}_F$ where we replace $\Sigma^{-1}$ with the Moore–Penrose pseudo-inverse of $\hat{\Sigma}$ could be as bad as random guessing and that, furthermore, the naive Bayes (NB) rule which ignores the correlation structure in $\Sigma$ typically outperforms $\hat{\Upsilon}_F$.

Continuing this line of inquiry, other independence rules (IR) have been proposed in Tibshirani et al. (2002) and Fan & Fan (2008) but these classifiers have two potentially major drawbacks, namely (1) the accumulation of error in estimating $\mu_1$ and $\mu_2$ and (2) mis-specification of the covariance matrix $\Sigma$. In particular Fan & Fan (2008) demonstrated that the noise accumulation in estimating $\mu_1$ and $\mu_2$ alone is sufficient to degrade the performance of IR classifiers, Shao et al. (2011) extended this result to show that the noise accumulation can also lead to the plug-in LDA classifier being, asymptotically, no better than random guessing even when $\Sigma$ is known, and finally Fan et al. (2012) showed that ignoring the correlations structure in $\Sigma$ prevents IR classifiers from achieving Bayes optimality. These results indicate the need for imposing sparsity conditions on the features and the important role of feature selection for mitigating the estimation errors associated with growing dimensionality.

To address the above limitations of IR classifiers, Shao et al. (2011) proposed thresholding of *both* $\Sigma$ and $\mu_2 - \mu_1$; this is similar to the motivation for regularized covariance estimators in Bickel & Levina (2009). In contrast, Cai & Liu (2011); Fan et al. (2012); Mai et al. (2012); Witten & Tibshirani (2011) and Cai & Zhang (2019) imposed sparsity conditions on the discriminant direction $\beta = \Sigma^{-1}(\mu_2 - \mu_1)$ and use penalized estimation approaches to recover $\beta$. Cai & Liu (2011) also noted two potential advantages to this approach, namely that the assumption of sparsity on $\beta$ is less restrictive than assuming sparsity for both $\Sigma$ (or $\Sigma^{-1}$) and $\mu_2 - \mu_1$, and secondly $\Upsilon_F$ only depends on $\Sigma^{-1}$ and $\mu_2 - \mu_1$ through their product $\Sigma^{-1}(\mu_2 - \mu_1)$ and thus consistent estimation of $\beta$ is sufficient. The sparsity assumption on $\beta$ leads to procedures and results that resemble those for high-dimensional linear regression even though the classification problem is generally not formulated in terms of a linear model. For example, Cai & Liu (2011) and Cai & Zhang (2019) considered a linear programming approach similar to the Dantzig selector (Candes & Tao, 2005) while Mai et al. (2012) studied a sparse discriminant analysis rule that used the Lasso (Tibshirani, 1996). Nevertheless it had been observed that, empirically, these approaches can lead to classification rules which select a larger number of features than necessary, and one possible explanation is that the correlation structure in $\Sigma$ also induced correlations among the entries for any estimate $\hat{\beta}$ of $\beta$.

In this paper we consider a different approach where we first perform dimension reduction on the feature vectors (using PCA) prior to classification (using LDA) in the dimensionally reduced space. As PCA is an important and ubiquitous pre-processing step in high-dimensional data analysis, there is a sizable number of work devoted to this approach. We refer to a generic classifier from this combination as lda ∘ pca. For example Section 9.1 of Jolliffe (2002) provides a detailed review of combining LDA with different variants of PCA in the low-dimensional setting while Niu et al. (2015) proposed the use of the reduced rank LDA together with class-conditional PCA in the high-dimensional settings. In terms of applications, lda ∘ pca is also used for faces and images recognition (Zhao et al., 1998; Belhumeur et al., 1997; Prasad et al., 2010), and recovering genetic patterns (Jombart et al., 2010)

While lda ∘ pca classifiers arise quite naturally, their theoretical properties in the high-dimensional setting remains an open problem. In particular their analysis requires possibly different techniques and assumptions compared to those based on direct estimation of the discriminant direction $\beta$. More specifically consistency results for $\beta$ are usually based on ideas from high-dimensional regression including assumptions on bounded and concentrated eigenvalues of $\Sigma$, see e.g., Bickel & Levina (2004); Fan & Fan (2008); Shao et al. (2011); Cai & Zhang (2019). However Wang & Fan (2017) noted that bounded eigenvalues are incompatible with the presence of strong signals (eigenvalues) in the data and might be problematic in fields such as genomics, economics and finance. These assumptions are nevertheless imposed mainly due to the limitation of quantifying the estimation error for $\Sigma$ in terms of the spectral norm difference for $\hat{\Sigma} - \Sigma$. In contrast the idealized setting for dimension reduction via PCA is when $\Sigma$ contains a small subset of signal eigenvalues that accounts for most of the variability in $\Sigma$. This idealized setting for PCA is also distinct from the idealized setting for graphical models wherein $\Sigma^{-1}$ is typically assumed to be sparse.

In summary, our contributions in this paper are as follows. We analyze the theoretical properties of a prototypical lda ∘ pca classifier under a spiked-covariance structure assumption – a widely-adopted covariance model for high-dimensional data – where $\Sigma$ contains a few large eigenvalues that are well-separated from the remaining (small) eigenvalues. In particular we show in Section 3 that lda ∘ pca is asymptotically Bayes-optimal as $n \to \infty$ and $n^{-1} \ln p \to 0$. This is, to the best of our knowledge, the first Bayes optimal consistency result for classification after performing dimension reduction via PCA. In Section 4 we demonstrate empirically, for both simulated and real data, that lda ∘ pca selects fewer features while also having error rates that are competitive with existing classifiers based on estimating the discriminant direction $\beta$. The theoretical and numerical results provide a clear example of the synergy linking dimension reduction with classification. Finally, in Section 5 we extend lda ∘ pca to classify data with (1) $K \geq 3$ classes or (2) two classes but with unequal covariance matrices or (3) feature vectors that are elliptical distributed but not necessarily multivariate normal.

## 2 Methodology

### 2.1 Notation and settings

For a vector $x \in \mathbb{R}^p$, the conventional $\ell_0$ quasi-norm and the $\ell_1$, $\ell_2$ and $\ell_\infty$ norms are denoted by $\|x\|_0$, $\|x\|_1$, $\|x\|_2$ and $\|x\|_\infty$, respectively. For $p \in \mathbb{N}$, we denote the set $\{1, ..., p\}$ by $[p]$. Given $x \in \mathbb{R}^p$ and a non-empty set $\mathcal{A} \subset [p]$, we write $x_\mathcal{A} = (x_j, j \in \mathcal{A})$ to denote the *column* vector obtained by keeping only the elements in $x$ whose indices belong to $\mathcal{A}$. The operation '∘', when applied to matrices, represents the Hadamard (entrywise) product. For $i \in [p]$, $\mathbf{e}_i^{(p)}$ is the $i$th standard basis vector of $\mathbb{R}^p$ and $\mathbf{1}_p$ is a vector in $\mathbb{R}^p$ whose elements are all 1; we also write $e_i$ and $\mathbf{1}$ when the choice of $p$ is clear from context and $\mathcal{I}_p$ stands for the identity matrix in $\mathbb{R}^{p \times p}$. For $x, y \in \mathbb{R}^p$, the standard Euclidean inner product between $x$ and $y$ is denoted as $\langle x, y \rangle := y^\top x$. For a matrix $M \in \mathbb{R}^{p \times q}$, the Frobenius norm and spectral norm of $M$ are written as $\|M\|_F$ and $\|M\|_2$, respectively. We will omit the subscript in $\|\cdot\|_2$ when it is clear from context that the spectral norm or $\ell_2$ norm is intended.

The two-to-infinity norm of $M$ is defined as

$$\|M\|_{2 \to \infty} := \sup_{\|x\|_2 = 1} \|Mx\|_\infty \equiv \max_{i \in [p]} \|M_i\|_2 \tag{2.1}$$

where $M_i$ represent the $i$th row of $M$. We note that $\|M\|_{2 \to \infty} \leq \|M\| \leq \|M\|_F$. Let $\mathbf{tr}(M)$ and $|M|$ denote the trace and determinant of a square matrix $M$. The effective rank of a square matrix $M$ is defined as $\mathbf{r}(M) := \mathbf{tr}(M)/\|M\|$; the effective rank of a matrix is a useful surrogate measure for its complexity, see e.g., Vershynin (2018). Let $\mathcal{O}(\cdot)$, $o(\cdot)$, $\Theta(\cdot)$ and $\Omega(\cdot)$ represent the standard big-O, little-o, big-Theta and big-Omega relationships. Finally, $\mathbb{1}\{\mathcal{A}\}$ and $|\mathcal{A}|$ stand for the indicator function of a set $\mathcal{A}$ and its cardinality, respectively.

In the subsequent discussion we shall generally assume, unless specified otherwise, that we have access to training set $\{X_{11}, \ldots, X_{1n_1}\}$ and $\{X_{21}, \ldots, X_{2n_2}\}$ whose elements are independently and identically distributed random vector from the $p$-variate distributions $\mathcal{N}_p(\mu_1, \Sigma)$ (class 1) and $\mathcal{N}_p(\mu_2, \Sigma)$ (class 2). The sample means for each class and the *pooled* sample covariance matrix are denoted as

$$\bar{X}_1 = \frac{1}{n_1} \sum_{j=1}^{n_1} X_{1j}, \qquad \bar{X}_2 = \frac{1}{n_2} \sum_{j=1}^{n_2} X_{2j}, \tag{2.2}$$

$$\hat{\Sigma} = \frac{1}{n} \Big( \sum_{j=1}^{n_1} (X_{1j} - \bar{X}_1)(X_{1j} - \bar{X}_1)^\top + \sum_{j=1}^{n_2} (X_{2j} - \bar{X}_2)(X_{2j} - \bar{X}_2)^\top \Big) \tag{2.3}$$

On numerous occasions we also need a variant of the sample covariance matrix where we replace the sample means with the true means. We denote this matrix as

$$\hat{\Sigma}_0 = \frac{1}{n} \Big( \sum_{j=1}^{n_1} (X_{1j} - \mu_1)(X_{1j} - \mu_1)^\top + \sum_{j=1}^{n_2} (X_{2j} - \mu_2)(X_{2j} - \mu_2)^\top \Big). \tag{2.4}$$

## 2.2 Linear discriminant analysis and whitening matrix

Let $\Sigma$ be a $p \times p$ positive definite matrix. The whitening matrix $\mathcal{W}$ is a linear transformation satisfying $\mathcal{W}\mathcal{W}^\top = \Sigma^{-1}$, and is generally used to decorrelate random variables and scale their variances to 1. The whitening transformation is unique only up to orthogonal transformation as $\mathcal{W}TT^\top\mathcal{W}^\top = \mathcal{W}\mathcal{W}$ for any $p \times p$ orthogonal matrix $T$; see Kessy et al. (2015) for a comparison between several common choices of whitening transformation. Hence, for this paper, we take $\mathcal{W} = \Sigma^{-1/2}$ as the *unique* positive semidefinite square root of $\Sigma^{-1}$. Note that $\Sigma^{-1/2}$ is the whitening transformation which minimizes the expected mean square error between the original data and the whitened data (Eldar & Oppenheim, 2003). Given $\mathcal{W}$, we define the *whitened* direction as $\zeta = \mathcal{W}(\mu_2 - \mu_1)$ and let $\mathcal{S}_\zeta = \{j : \zeta_j \neq 0\}$. We referred to the elements of $\mathcal{S}_\zeta$ as the whitened coordinates or variables. The Fisher linear discriminant rule is then equivalent to

$$\Upsilon_F(\mathbf{Z}) = \begin{cases} 1 & \text{if } \zeta^\top\left(\mathcal{W}\left(\mathbf{Z} - \frac{\mu_1+\mu_2}{2}\right)\right) = \zeta_{\mathcal{S}_\zeta}^\top\left(\mathcal{W}\left(\mathbf{Z} - \frac{\mu_1+\mu_2}{2}\right)\right)_{\mathcal{S}_\zeta} \leq \ln\frac{\pi_1}{1-\pi_1}, \\ 2 & \text{otherwise,} \end{cases} \tag{2.5}$$

where $\mathbf{Z} \sim \pi_1\mathcal{N}_p(\mu_1, \Sigma) + (1-\pi_1)\mathcal{N}_p(\mu_2, \Sigma)$. Recall that, for a vector $\xi$ and a set of indices $\mathcal{A}$, the vector $\xi_{\mathcal{A}}$ is obtained from $\xi$ by keeping only the elements indexed by $\mathcal{A}$. Since $\beta = \Sigma^{-1}(\mu_2 - \mu_1)$ is the Bayes direction, a significant portion of previous research is devoted to recovering $\beta$ and the discriminative set $\mathcal{S}_\beta = \{j : \beta_j \neq 0\}$. We refer to the elements of $\mathcal{S}_\beta$ as the discriminative coordinates or variables. We emphasize that in general the discriminative set and whitened set are not the same, except for when $\Sigma$ has certain special structures, e.g., $\Sigma$ being diagonal.

If $p \ll n$ then the empirical whitening matrix $\hat{\mathcal{W}} = \hat{\Sigma}^{-1/2}$ is well-defined and the empirical whitened variables $\hat{\zeta} = \hat{\mathcal{W}}(\bar{X}_2 - \bar{X}_1)$ are approximately decorrelated in the plug-in LDA rule, i.e., by the law of large numbers, $\hat{\mathcal{W}}\Sigma\hat{\mathcal{W}} \to \mathcal{I}_p$ and hence $\text{Var}[\hat{\zeta}] \approx c\mathcal{I}_p$ for some constant $c > 0$. In contrast, the coordinates of the estimated discriminant direction $\hat{\beta}$ are not decorrelated since $\text{Var}[\hat{\beta}] \approx c\Sigma^{-1}$. It is thus easier to quantify the impact of any arbitrary *whitened* feature toward the classification accuracy than to quantify the impact of an arbitrary *raw* feature.

## 2.3 Combining LDA and PCA

Eq. (2.5) can be viewed under the framework of first performing dimension reduction and then doing classification in the lower-dimensional spaces with the main focus being that thresholding of the discriminant direction corresponds to dropping a subset of the *raw* features while thresholding of the whitened direction corresponds to dropping a subset of the *whitened* features. We now discuss the estimation of the whitening matrix $\mathcal{W}$.

If $n \asymp p$ then the sample covariance matrix $\hat{\Sigma}$ is often-times ill-conditioned, and is furthermore singular when $n < p$. The estimation of either the precision matrix $\Sigma^{-1}$ or the whitening matrix $\mathcal{W}$ is thus challenging in these regimes, especially since $\Sigma^{-1}$ and $\mathcal{W}$ both contains $O(p^2)$ entries. An universal approach to address this difficulty is to reduce the number of parameters needed for estimating $\Sigma$, for example by assuming that $\Sigma$ has a parametric form with $o(p^2)$ parameters, or by introducing regularization terms to induce certain structures in the estimate $\hat{\Sigma}$.

The use of regularized estimators for covariance matrices are popular in high-dimensional classification, see e.g., Bickel & Levina (2009); Shao et al. (2011), and theoretical analysis for these estimators are usually based on assumptions about the sparsity of either $\Sigma$ or $\Sigma^{-1}$ and then bounding the estimation error in terms of the spectral norm differences $\|\hat{\Sigma}^{-1} - \Sigma^{-1}\|$ or $\|\hat{\Sigma} - \Sigma\|$. There are, however, two potential drawbacks to this approach. Firstly, quantifying the estimation error in terms of the spectral norm can be quite loose and the resulting bounds might not capture the difference in geometry between the eigenspaces of $\Sigma$ and that of the perturbed matrix $\hat{\Sigma}$ (e.g., Johnstone (2001) observed that the first few largest eigenvalues of the sample covariance matrix $\hat{\Sigma}$ are always larger than those for $\Sigma$ in the high-dimensional setting). Secondly, while it is not always appropriate to assume sparsity of $\Sigma$ or $\Sigma^{-1}$, Cai & Liu (2011) noted that some form of sparsity is needed to guarantee consistent estimation of $\Sigma$ or $\Sigma^{-1}$ under spectral norm error.

In this paper we assume a different structure for $\Sigma$, namely that $\Sigma$ is a spiked-covariance matrix with a few leading eigenvalues (the spike) that are well-separated from the remaining eigenvalues (the bulk). More specifically, we assume that $\Sigma$ satisfies the following condition.

*Assumption* 1 (Spiked covariance matrix). Let $\mathbf{u}_1, \dots, \mathbf{u}_d$ be orthonormal vectors in $\mathbb{R}^p$ and assume that the covariance matrix $\Sigma$ for the $p$-variate distributions $\mathcal{N}_p(\mu_1, \Sigma)$ and $\mathcal{N}_p(\mu_2, \Sigma)$ is of the form

$$\Sigma = \sum_{k=1}^{d} \lambda_k \mathbf{u}_k \mathbf{u}_k^\top + \sigma^2 \mathcal{I}_p = \mathcal{U}\Lambda\mathcal{U}^\top + \sigma^2 \mathcal{I}_p. \tag{2.6}$$

Here $\Lambda = \mathrm{diag}(\lambda_k)$ is a $d \times d$ diagonal matrix, $\mathcal{U} = (\mathbf{u}_k)$, $k \in [d]$ is a $p \times d$ matrix with orthonormal columns, and $\mathcal{I}_p$ is the identity matrix. We assume implicitly that $\lambda_1 \geq \cdots \geq \lambda_d > 0$, $\sigma > 0$ and $d \ll p$.

Covariance matrices with spiked structures have been studied extensively in the high-dimensional statistics literature, see e.g., Johnstone (2001) and Chapter 11 of Yao et al. (2015), and there is a significant number of results for consistent estimation of $\mathcal{U}$ under the assumption that the support of $\mathcal{U}$ is sparse, e.g., either that the number of non-zero rows of $\mathcal{U}$ is small compared to $p$ or that the $\ell_q$ quasi-norm, for some $q \in [0, 1]$, of the columns of $\mathcal{U}$ are bounded. The case when $q = 0$ and $q > 0$ correspond to "hard" and "soft" sparsity constraints, respectively; see for example Birnbaum et al. (2012); Berthet & Rigollet (2012); Vu & Lei (2012); Cai et al. (2013) and the references therein. In contrasts to the above cited results, in this paper we do not impose sparsity conditions on $\mathcal{U}$ but instead assume that $\mathcal{U}$ have bounded coherence, namely that the maximum $\ell_2$ norm of the rows of $\mathcal{U}$ are of order $O(p^{-1/2})$; see Assumption 4 for a precise statement. The resulting matrix $\Sigma$ will no longer be sparse. The main rationale for assuming bounded coherence is that the spiked eigenvalues $\Lambda$ can grow linearly with $p$ while still guaranteeing bounded variance in $\Sigma$. There is thus a large gap between the spiked eigenvalues and the bulk eigenvalues, and this justifies the use of PCA as a pre-processing step.

If $\Sigma$ satisfies Condition 1 then the whitening matrix is given by

$$\mathcal{W} = \mathcal{U}\mathcal{D}\mathcal{U}^\top + \frac{1}{\sigma}(\mathcal{I}_p - \mathcal{U}\mathcal{U}^\top) \tag{2.7}$$

where $\mathcal{D} = \mathrm{diag}(\eta_k)$ is a $d \times d$ diagonal matrix with diagonal entries $\eta_k = (\lambda_k + \sigma^2)^{-1/2}$. The above form for $\mathcal{W}$ suggests the following estimation procedure.

1. Extract the $d$ largest eigenvalues and corresponding eigenvectors of the pooled sample covariance matrix $\hat{\Sigma}$. Let $\hat{\Lambda}$ denote the diagonal matrix of these $d$ largest eigenvalues and $\hat{\mathcal{U}}$ denote the $p \times d$ orthogonal matrix whose columns are the corresponding eigenvectors.

2. Estimate the non-spiked eigenvalues by

$$\hat{\sigma}^2 = \frac{\mathrm{tr}(\hat{\Sigma}) - \mathrm{tr}(\hat{\Lambda})}{p - d}. \tag{2.8}$$

3. Let $\hat{\mathcal{D}} = (\hat{\Lambda} + \hat{\sigma}^2 \mathcal{I}_d)^{-1/2}$ and estimate $\mathcal{W}$ by $\hat{\mathcal{W}} = \hat{\mathcal{U}}\hat{\mathcal{D}}\hat{\mathcal{U}}^\top + \hat{\sigma}^{-1}(\mathcal{I}_p - \hat{\mathcal{U}}\hat{\mathcal{U}}^\top)$.

Although the sample covariance matrix $\hat{\Sigma}$ is generally a poor estimate of $\Sigma$ when $n \ll p$, the eigenvectors $\hat{\mathcal{U}}$ corresponding to the $d$ largest eigenvalues of $\hat{\Sigma}$ are nevertheless accurate estimates of $\mathcal{U}$. In particular, Fan et al. (2018) and Cape et al. (2019) provide uniform error bound for $\min_T \|\hat{\mathcal{U}}T - \mathcal{U}\|_{2\to\infty}$ where the minimum is taken over all $d \times d$ orthogonal transformations $T$; see also Theorem 1 of the current paper. We can thus transform $\hat{\mathcal{U}}$ by an orthogonal transformation $T$ so that the resulting (transformed) rows of $\hat{\mathcal{U}}$ are uniformly close to the corresponding rows of $\mathcal{U}$.

Given the above estimate for $\mathcal{W}$, we then have the prototypical lda∘pca classifier in Algorithm 1. Algorithm 1 contains two tuning parameters, namely $d$, the number of principal components in the PCA step and $s$, the number of coordinates of estimated whitened direction $\hat{\zeta}$ that we preserve. The choices for $d$ and $s$ correspond to the number of spiked eigenvalues in $\Sigma$ and the sparsity level of the whitened direction $\zeta$. We show in

Section 3 that, under certain mild conditions, one can consistently estimate $s$ using Eq. (3.6). Consistent estimation of $d$ can be obtained using results in Bai & Ng (2002); Alessi et al. (2010); Hallin & Liška (2007) among others. Algorithm 1 thus yields a classifier that is adaptive with respect to both $d$ and $s$.

---

**Algorithm 1:** lda ∘ pca decision rule

---

**Input:** $\bar{X}_1$, $\bar{X}_2$, $\hat{\Sigma}$ and the test sample $\mathbf{Z}$
**Output:** $\hat{\Upsilon}_{\mathrm{lda \circ pca}}(\mathbf{Z})$
**Algorithm**

  // Step 1: Perform PCA on the feature vectors for the training data (standard PCA approach)

  // Step 2: Extract the $d$ largest principal components and obtain $\hat{\mathcal{W}} = \hat{\mathcal{U}}\hat{\mathcal{D}}\hat{\mathcal{U}}^{\top} + \hat{\sigma}^{-1}(\mathcal{I}_p - \hat{\mathcal{U}}\hat{\mathcal{U}}^{\top})$.

  // Step 3: Take $\tilde{X}_i = \hat{\mathcal{W}}\bar{X}_i$, $i = 1, 2$ and $\tilde{X}_a = 0.5(\tilde{X}_1 + \tilde{X}_2)$. Let $\hat{\zeta} = \tilde{X}_2 - \tilde{X}_1$ and form the indices set $\hat{\mathcal{S}}$ by selecting the $s$ largest coordinates of $\hat{\zeta}$ in modulus.

  // Step 4: Given the test sample $\mathbf{Z}$, take $\tilde{\mathbf{Z}} = \hat{\mathcal{W}}\mathbf{Z}$ and plug the sub-vector of $\hat{\zeta}$, $\tilde{\mathbf{Z}}$ and $\tilde{X}_a$ corresponding to the indices in $\hat{\mathcal{S}}$ into the Fisher discriminant rule, i.e.,

$$\hat{\Upsilon}_{\mathrm{lda \circ pca}}(\mathbf{Z}) = \begin{cases} 1 & \text{if } \hat{\zeta}_{\hat{\mathcal{S}}}^{\top}\left[\tilde{\mathbf{Z}} - \tilde{X}_a\right]_{\hat{\mathcal{S}}} \leq \ln \frac{n_1}{n_2}, \\ 2 & \text{otherwise} \end{cases} \tag{2.9}$$

---

## 2.4 Related works

### 2.4.1 Whitening matrix and spiked covariance

We first discuss the relationship between Algorithm 1 and two other relevant classifiers, namely the features annealed independence rule (FAIR) of Fan & Fan (2008) and the LDA with CAT scores (CAT-LDA) of Zuber & Strimmer (2009). FAIR is the independence rule applied to the features prescreened by the two-sample $t$ test while CAT-LDA scores decorrelate the $t$ statistics by first whitening the data using the *sample correlation* matrix $\mathrm{diag}(\hat{\Sigma})^{-1/2}\,\hat{\Sigma}\,\mathrm{diag}(\hat{\Sigma})^{-1/2}$; CAT-LDA is motivated by the empirical observation that accounting for correlations is essential in the analysis of proteomic and metabolic data.

If there are no correlation or if the correlations are negligible, i.e., when $\Sigma$ is diagonal or approximately diagonal, then Algorithm 1 is essentially equivalent to both FAIR and CAT-LDA. However, when the correlations are not negligible, the performance of IR classifiers such as FAIR degrades significantly due to mis-specification of the covariance structure (Fan et al., 2012). Both Algorithm 1 and CAT-LDA apply a whitening transformation before doing LDA but the choice of whitening matrices are different between the two procedures. In particular CAT-LDA uses the asymmetric whitening transformation $\mathrm{Corr}(\Sigma)^{-1/2}\mathrm{diag}(\Sigma)^{-1/2}$ where $\mathrm{Corr}(\Sigma)$ is the matrix of correlations. The choice of whitening transformation is not important if $\Sigma$ is known or if we are in the low-dimensional setting where $p \ll n$ for which consistent estimation of $\Sigma$ is straightforward. It is only when $p \asymp n$ or $p \gg n$ when the different regularization strategies used in CAT-LDA and Algorithm 1 lead to possibly different behaviors for the resulting classifiers; see for example the numerical comparisons given in Section 4 of the current paper. Finally we view the theoretical results in Section 3 as not only providing justification for Algorithm 1 but also serve as examples of theoretical analysis that can be extended to other classifiers in which dimension reduction is done prior to performing LDA. Indeed, to the best of our knowledge, there are no theoretical guarantees for the error rate for the CAT-LDA classifier in the high-dimensional setting.

Finally we note that our paper is not the first work to study high-dimensional classification under spiked covariance matrices of the form in Condition 1. In particular Sifaou et al. (2020a) also analyzed the performance of LDA under spiked covariance structures; *however*, their theoretical results are premised under different settings and framework from ours and most importantly they *do not* show that their classification rule achieves the Bayes error rate; see Remark 4 for detailed comparisons of the assumptions and theoretical

results in our paper against that of Sifaou et al. (2020a). With a different focus, Hao et al. (2015) examined the effects of spiked eigenvalues when trying to find an orthogonal transformation $T$ of the discriminant direction $\beta$ so that $T\beta$ is sparse. More specifically they show that if $\Sigma$ is known then $T$ can be computed using the eigen-decomposition of the matrix $\Sigma_{\text{rot}}$ where

$$\Sigma_{\text{rot}} = \Sigma + \gamma \Delta\mu \Delta\mu^\top, \qquad \text{for a given } \gamma > 0 \tag{2.10}$$

where $\Delta\mu = \mu_2 - \mu_1$ is the vector of mean differences between the two classes. Let $\mathcal{U}_{\text{rot},m}$ be the $p \times m$ matrix whose columns consist of the orthonormal eigenvectors of $\Sigma_{\text{rot}}$ corresponding to the $m$ largest eigenvalues. When $m = p$ then $\mathcal{U}_{\text{rot},p}$ diagonalizes $\Sigma_{\text{rot}}$ and Hao et al. (2015) showed that $\mathcal{U}_{\text{rot},p}$ further sparsifies $\beta$ in that $\|\mathcal{U}_{\text{rot},p}^\top \beta\|_0 \leq d + 1$ whenever $\Sigma$ satisfies the spiked covariance assumption in Assumption 1, and hence it might be beneficial to rotate the data before performing classification.

When $\Sigma$ is unknown Hao et al. (2015) propose the following procedure for estimating $\mathcal{U}_{\text{rot},m}$ for some choice of $m \leq \min\{n, p\}$; here $\gamma > 0$ is a user-specified parameter.

---

**Algorithm 2:** Rotation as a preprocessing in LDA

**Algorithm**

> // Step 1: Perform PCA on $\hat{\Sigma}_{\text{rot}} = \hat{\Sigma} + \gamma(\bar{X}_2 - \bar{X}_1)(\bar{X}_2 - \bar{X}_1)^\top$ to extract the $m$ largest principal components. Let $\hat{\mathcal{U}}_{\text{rot},m}$ be the resulting $p \times m$ matrix.
> // Step 2: Rotate the training set to $\{\hat{\mathcal{U}}_{\text{rot},m}^\top X_{i1}, \ldots, \hat{\mathcal{U}}_{rot,m}^\top X_{in_i}\}$ where $\hat{\mathcal{U}}_{\text{rot},m}^\top X_{ij} \in \mathbb{R}^m$ for $i \in \{1, 2\}$ and $j \in \{1, 2, \ldots, n_i\}$.
> // Step 3: Apply some discriminant direction based LDA method such as Fan et al. (2012); Cai & Liu (2011); Mai et al. (2015) on the rotated data set $\{\hat{\mathcal{U}}_{\text{rot},m}^\top X_{ij}\}$.

---

We will use this pre-processing step in some of our numerical experiments in Section 4. Nevertheless we emphasize that the theoretical results of Hao et al. (2015) assume $\Sigma$ is known as they are mainly concerned with the analysis of different approaches for sparsifying $\beta$, i.e., they explicitly chose not to address the important issue of how the estimation error for $\hat{\mathcal{U}}_{\text{rot},m}$ impacts the classification accuracy.

### 2.4.2 Connection to high-dimensional sparse LDA

Let $\mathbf{X}$ be the $(n_1 + n_2) \times p$ matrix whose rows are the $\{X_{ij}\}$ and $\mathcal{Y} \in \mathbb{R}^n$ be the vector whose first $n_1$ elements are set to $-(n/n_1)$ and the remaining $n_2$ elements are set to $n/n_2$. Mai et al. (2012) reframed LDA in high-dimension as the solution to a Lasso-type problem (we have omitted the intercept term for simplicity of presentation)

$$\hat{\beta} = \underset{\beta \in \mathbb{R}^p}{\arg\min} \ \frac{1}{2}\|\mathcal{Y} - \mathbf{X}\beta\|^2 + \lambda\|\beta\|_1 \tag{2.11}$$

for some $\lambda > 0$. In a similar spirit to Eq. (2.11), we might, *conceptually*, also reformulate Algorithm 1 as a (general) Lasso-type problem (Tibshirani & Taylor (2011))

$$\hat{\beta} = \underset{\beta \in \mathbb{R}^p}{\arg\min} \ \frac{1}{2}\|\mathcal{Y} - \mathbf{X}\beta\|^2 + \lambda\|\Psi\beta\|_1 \tag{2.12}$$

for some $\lambda > 0$, where $\Psi = \Sigma^{1/2}$ (once again omitting the intercept for simplicity). As $\Sigma$ is invertible, Eq. (2.12) is equivalent to first solving

$$\hat{\zeta} = \underset{\zeta \in \mathbb{R}^p}{\arg\min} \ \frac{1}{2}\|\mathcal{Y} - \tilde{\mathbf{X}}\zeta\|^2 + \lambda\|\zeta\|_1 \tag{2.13}$$

where $\tilde{\mathbf{X}} = \mathbf{X}\Sigma^{-1/2}$, and then setting $\hat{\beta} = \Sigma^{-1/2}\hat{\zeta}$. Comparing Eqs. (2.12) and (2.13) against Eq. (2.11), one could argue that the main difference between lda ∘ pca in Algorithm 1 and the lassoed LDA classifier of Mai et al. (2012) is due to the different choice of the predictor variables $\mathbf{X}$ vs $\tilde{\mathbf{X}}$ in the Lasso regression.

This argument, however, overlooks the important fact that the transformation $\Sigma^{-1/2}$ needs to be estimated in Algorithm 1 while for the general Lassso in Eq. (2.12), the matrix $\Psi$ is specified *a priori* and thus free from any estimation error. Indeed, while there have been significant efforts devoted to understanding the solution path of Eq. (2.12) in the high-dimensional setting – see, for example, Duan et al. (2016); Tibshirani & Taylor (2011) among others – its behavior when $\Psi$ is determined empirically from the data is yet to be theoretically investigated.

Next we note that, instead of Eq. (2.11), one can also estimate $\zeta$ via an $\ell_1$ optimization problem similar to that of the Dantzig selector (Candes & Tao, 2005; Cai & Liu, 2011; Cai & Zhang, 2019), namely

$$\hat{\zeta} = \arg\min_{\zeta \in \mathbb{R}^p} \left\{ \|\zeta\|_1 \text{ subject to } \|\hat{\Sigma}^{1/2}\zeta - (\bar{X}_2 - \bar{X}_1)\|_\infty \leq \lambda \right\}. \tag{2.14}$$

The main difference between Algorithm 1 and Eq. (2.14) is that Algorithm 1 first whitens the feature vectors using an empirical estimate of the covariance matrix followed by feature selection on the whitened data (see Eq. (2.13)). In contrast, Eq. (2.14) attempts to find a $\zeta$ for which its transformation $\hat{\Sigma}^{1/2}\zeta$ is most similar to $\bar{X}_2 - \bar{X}_1$. While the approaches underlying Algorithm 1 and Eq. (2.14) are quite similar, we believe that Algorithm 1 provides a more direct link between PCA and LDA for high-dimensional classification as it is particularly suitable for high-dimensional data generated from a (possibly low-dimenisonal) factor model (where the covariance matrix will now have a spiked structure with leading eigenvalues that grow with the dimension $p$). See Fan et al. (2021) for further discussion of factor models and their applications to statistics and machine learning.

Although Algorithm 1 and Eq. (2.11) share a conceptual link in optimization formulation, they differ significantly in algorithmic structure, leading to distinct computational profiles. We now compare the computational complexity of lda∘pca and Lassoed LDA. For our method, the dominant cost comes from computing the top $\bar{d}$ singular values of the centered data matrix $\mathbf{X}$, with cost $O(np\bar{d})$, where $\bar{d}$ upper-bounds the rank $d$; see Halko et al. (2011) and Feng & Yu (2023) for more details. In contrast, Lassoed LDA solves an $\ell_1$-penalized least squares problem with complexity $O(npT)$, where $T$ is the number of iterations until convergence. The iteration count $T$ can be substantial in practice, especially when regularization is weak or $\mathbf{X}$ is ill-conditioned as often occurs in settings with low-rank or highly correlated features. Consequently, our method provides substantial computational advantages in high-dimensional latent variable models, while maintaining competitive classification accuracy and working in a reduced-dimensional space (see Section 4).

Finally we conclude this section by comparing Algorithm 1 with the principal component classifiers proposed in Bing et al. (2024); Bing & Wegkamp (2023). In particular both Bing & Wegkamp (2023) and Bing et al. (2024) assumed that the discriminant direction $\beta$ (and the whitening direction $\zeta$ as in our work) lies entirely within the column space spanned by the $d$ leading principal components $\mathcal{U}$. However, as noted by Jolliffe (1982), the principal components associated with small eigenvalues can be just as important as those associated with large eigenvalues in real data analysis (see also the simulation settings for Model 1 and Model 3 in Section 4). By working with the whitened data, Algorithm 1 also takes into account the *non-leading* principal components and furthermore, by performing feature selection in the whitened space, avoid the need to specify *a priori* the low-dimensional subspaces containing $\beta$ and/or $\zeta$.

## 3 Theoretical properties

In this section, we derive the theoretical properties of the lda∘pca classifier in Algorithm 1 for the case where the feature vectors $X$ are sampled from a mixture of two multivariate Gaussians. Extensions of these results to the case where $X$ is a mixture of $K \geq 2$ elliptical, but not necessarily multivariate normal, distributions are discussed in Section 7.3 and Section 5.1.

If $X \sim \pi_1 \mathcal{N}_p(\mu_1, \Sigma) + (1 - \pi_1)\mathcal{N}_p(\mu_2, \Sigma)$ then the Fisher's rule $\Upsilon_F$ has error rate

$$R_F = \pi_1 \Phi\left(-\frac{1}{2}\|\zeta\| + \|\zeta\|^{-1} \ln \frac{1 - \pi_1}{\pi_1}\right) + (1 - \pi_1)\Phi\left(-\frac{1}{2}\|\zeta\| - \|\zeta\|^{-1} \ln \frac{1 - \pi_1}{\pi_1}\right). \tag{3.1}$$

Here $\Phi$ is the cumulative distribution function for $\mathcal{N}(0, 1)$ and $\zeta = \Sigma^{-1/2}(\mu_2 - \mu_1)$; see e.g., Ripley (1996, Section 2.1). We note that $R_F$ is the smallest mis-classification error achievable by any classifier and,

furthermore, is monotone decreasing as $\|\zeta\|$ increases. If $\|\zeta\| \to 0$ then $R_F \to \min\{\pi_1, 1 - \pi_1\}$ and $\Upsilon_F$ is no better than assigning every data point to the most prevalent class, while if $\|\zeta\| \to \infty$ then $R_F \to 0$ and $\Upsilon_F$ achieves perfect accuracy. The cases where $R_F \to 0$ or $R_F \to \min\{\pi_1, 1 - \pi_1\}$ are, theoretically, uninteresting and hence in this paper we only focus on the case where $0 < \|\zeta\| < \infty$. We therefore make the following assumption.

*Assumption* 2. Let $S_\zeta$ denote the set of indices $i$ for which $\zeta_i \neq 0$. Also let $\mathcal{C}_0 > 0$, $M > 0$ and $\mathcal{C}_\zeta > 0$ be constants not depending on $p$ such that $s_0 := |\mathcal{S}_\zeta| \leq M$ and

$$\min_{j \in \mathcal{S}_\zeta} |\zeta_j| \geq \mathcal{C}_0, \qquad \max\left\{\|\Sigma^{-1/2}\mu_1\|, \|\Sigma^{-1/2}\mu_2\|\right\} \leq \mathcal{C}_\zeta.$$

In addition, we assume that the number of spikes $d$ in the spiked covariance model (see Assumption 1) is fixed and does not grow with $p$ or $n$.

*Remark* 1. Assumption 2 implies $0 < \mathcal{C}_0\sqrt{s_0} \leq \|\zeta\| \leq 2\mathcal{C}_\zeta < \infty$. We emphasize that sparsity is imposed on the *whitened* direction so that only a few *transformed* features contribute to classification outcome; similar assumptions can be found in Silin & Fan (2022). To further explain this condition, we note that sparsity of the *discriminant* direction $\beta = \Sigma^{-1}(\mu_1 - \mu_2)$ also implies that only a small subset of the raw covariates affects the response (as the classification boundary for a feature vector $\boldsymbol{x}$ is given by $\mathbf{1}\{\boldsymbol{x}^\top\beta > a\}$ for some $a \in \mathbb{R}$). However, as noted by Zhu & Bradic (2018) and Hall et al. (2014), sparsity of $\beta$ is incompatible with many real data application such as genome-wide gene expression profiling where all genes are believed to play a role in disease markers, or analysis of micro-array data to identify leukemia or colon/prostate cancer. In contrast sparsity of the whitened direction $\zeta = \Sigma^{-1/2}(\mu_1 - \mu_2)$ still allows for $\beta = \Sigma^{-1/2}\zeta$ to be non-sparse, thereby circumventing the issue described above. Lastly, throughout our theoretical results and proofs, we assume that the sparsity level $s_0$ is fixed and independent of the sample size $n$, though it may be arbitrary. The case where $s_0$ grows with $n$ can be handled *mutatis mutandis*. However, it requires more careful control of technical arguments and is therefore left to the interested reader.

Our theoretical results are large-sample results in which the sample sizes $n_1$ and $n_2$ for the training data increase as the dimension $p$ increases, and thus our next assumption specifies the asymptotic relationships between these quantities and the eigenvalues of $\Sigma$.

*Assumption* 3. Let $\sigma > 0$ be fixed and suppose that

$$\frac{n_1}{n_2} = \Theta(1), \quad \ln p = o(n).$$

Furthermore, the spiked eigenvalues $\lambda_1, \ldots, \lambda_d$ of $\Sigma$ satisfy

$$\lambda_k = \Theta(p), \qquad \text{for all } k \in [d].$$

If $\Sigma$ satisfies Assumption 3 then $\mathbf{tr}(\Sigma) = \Theta(p)$ and $\boldsymbol{r}(\Sigma) = \text{tr}(\Sigma)/\|\Sigma\| = \mathcal{O}(1)$. Recall that $\boldsymbol{r}(\Sigma)$ is the *effective* rank of $\Sigma$. Assumption 3 also implies that the spiked eigenvalues of $\Sigma$ are unbounded as $p$ increases and this assumption distinguishes our theoretical results from existing results in the literature wherein it is generally assumed that the eigenvalues of $\Sigma$ are bounded; see Bickel & Levina (2004); Fan & Fan (2008); Shao et al. (2011); Cai & Zhang (2019) for a few examples of results under the bounded eigenvalues assumption. While the bounded eigenvalues assumption is prevalent, it can also be problematic for high-dimensional data as it ignores the strong signals present in many real data applications; see Fan et al. (2013) and Wang & Fan (2017) for further discussions of this issue. Indeed, a standard heuristic for PCA is to keep the $d$ largest principal components that explains 90% or 95% of the variability in the data, and hence if $p$ is large and $\Sigma$ has bounded eigenvalues then one has to choose $d = \Theta(p)$ which is inconsistent with the use of PCA as a dimension reduction procedure.

If $\Sigma$ has a spiked eigenvalue structure as in Assumption 1 then $\mathcal{W}$ is given by

$$\mathcal{W} = \Sigma^{-1/2} = \mathcal{U}(\Lambda + \sigma^2\mathcal{I})^{-1/2}\mathcal{U}^\top + \sigma^{-1}(\mathcal{I} - \mathcal{U}\mathcal{U})^\top \tag{3.2}$$

and a natural estimate for $\mathcal{W}$ is

$$\hat{\mathcal{W}} = \hat{\mathcal{U}}(\hat{\Lambda} + \hat{\sigma}^2\mathcal{I})^{-1/2}\hat{\mathcal{U}}^\top + \hat{\sigma}^{-1}(\mathcal{I} - \hat{\mathcal{U}}\hat{\mathcal{U}}^\top) \tag{3.3}$$

where $\hat{\Lambda}$ is the diagonal matrix containing the $d$ largest eigenvalues of the *pooled* covariance matrix, $\hat{\mathcal{U}}$ is the $p \times d$ orthonormal matrix whose columns are the corresponding eigenvectors, and $\hat{\sigma}^2$ is as defined in Eq. (2.8). We now make the following assumption on $\mathcal{U}$.

*Assumption* 4 (Bounded Coherence). There is a constant $\mathcal{C}_{\mathcal{U}} \geq 1$ independent of $n$ and $p$ such that

$$\|\mathcal{U}\|_{2 \to \infty} \leq \frac{\mathcal{C}_{\mathcal{U}} \sqrt{d}}{\sqrt{p}}.$$

*Remark* 2. The bounded coherence assumption appears frequently in statistical inference for matrix-valued data. More specifically, as every column in $\mathcal{U}$ has $\ell_2$ norm equal to 1, the rows $\mathcal{U}$ have, on average, $\ell_2$ norm of order $\sqrt{d/p}$. Assumption 4 then guarantees that the *maximum* $\ell_2$ norm of the rows of $\mathcal{U}$ is also of order $\sqrt{d/p}$. For more discussions about the bounded coherence assumption in the context of matrix completion, covariance matrix estimation, and random matrix theory, see Candès & Recht (2009); Fan et al. (2018); Rudelson & Vershynin (2016); Bloemendal et al. (2014) among others. Finally we note that if $\mathcal{U}$ satisfies the bounded coherence assumption with constant $\mathcal{C}_{\mathcal{U}}$ then

$$\Sigma_{ii} \leq \frac{\mathcal{C}_{\mathcal{U}}^2 \lambda_1 d}{p} + \sigma^2, \qquad \text{for all } i \in [p],$$

which together with Assumption 3 implies $\max_{i \in [p]} \Sigma_{ii} = \mathcal{O}(1)$. In summary, Assumption 4 allows for the spiked eigenvalues $\lambda_1, \ldots, \lambda_d$ of the covariance matrix $\Sigma$ to grow linearly with $p$ while also guaranteeing that the entries of $\Sigma$ remains bounded, i.e., each variable in $X_{ij}$ has a finite variance.

We next state a result on the estimation accuracy of $\hat{\mathcal{U}}$. This result is a slight extension of an earlier result in Cape et al. (2019). More specifically, Cape et al. (2019) assume $\mathbb{E}[\mathbf{X}] = 0$ and hence the sample covariance matrix is simply $\frac{1}{n}\mathbf{X}^\top \mathbf{X}$. In this paper we used the *pooled* sampled covariance matrix which requires first centering the feature vectors by the sample means of each class.

**Theorem 1.** *Let $\mathbf{X}$ be a $n \times p$ matrix where the rows $X_1, \ldots, X_n$ are i.i.d samples from $\pi_1 \mathcal{N}(\mu_1, \Sigma) + (1 - \pi_1)\mathcal{N}(\mu_2, \Sigma)$ and $\Sigma$ satisfies Assumption 1, Assumption 3, and Assumption 4. Let $\hat{\mathcal{U}}$ be the matrix of eigenvectors corresponding to the $d$ largest eigenvalues of the pooled sample covariance matrix $\hat{\Sigma}$. Then there exists a $d \times d$ orthogonal matrix $\Xi_{\mathcal{U}}$ and a constant $C > 0$ such that with probability at least $1 - \mathcal{O}(p^{-2})$,*

$$\|\hat{\mathcal{U}} - \mathcal{U}\Xi_{\mathcal{U}}\|_{2 \to \infty} \leq C \sqrt{\frac{d^3 \ln p}{np}}. \tag{3.4}$$

Note that, for simplicity, we assume in Theorem 1 as well in the subsequent part of this paper that $d$, the number of spiked eigenvalues, is known. If $d$ is unknown then it can be *consistently* estimated using the ratio between consecutive eigenvalues, similar to the procedures in Ahn & Horenstein (2013). More specifically, from Assumption 3 and Lemma 1 we have with high probabliity that $\hat{\lambda}_k = \Theta(p)$ for $k \leq d$ and $\hat{\lambda}_k = O(\sigma^2)$ for $k \geq d+1$. Here $\hat{\lambda}_k$ are the eigenvalues of $\hat{\Sigma}$. There thus exists a significant gap between $\hat{\lambda}_{d-1}/\hat{\lambda}_d = O(1)$ and $\hat{\lambda}_d/\hat{\lambda}_{d+1} = \Omega(p)$, and hence, letting $\hat{d}$ be the smallest index $k$ for which $\hat{\lambda}_k/\hat{\lambda}_{k+1} = \Omega(\ln p)$, we have $\hat{d} = d$ asymptotically almost surely.

We now analyze the classification accuracy of lda ∘ pca. Recall that the main idea behind lda ∘ pca is that we first construct an estimate $\hat{\zeta}$ for the whitened direction $\zeta$ and then project our *whitened* data onto the $d$ largest coordinates, in magnitude, of $\hat{\zeta}$ (see Algorithm 1). By Eq. (3.1), the Bayes error rate $R_F$ is a monotone decreasing function of $\|\zeta\|$ and hence, to achieve $R_F$ it is only necessary to recover the indices in $S_\zeta = \{i : \zeta_i \neq 0\}$. In summary the error rate for lda ∘ pca converges to $R_F$ provided we can (1) bound the error $\hat{\zeta} - \zeta$ and (2) show that thresholding $\hat{\zeta}$ perfectly recovers $S_\zeta$.

**Theorem 2.** *Under Assumptions 1-4, there exists a constant $C > 0$ such that with probability at least $1 - \mathcal{O}(p^{-2})$,*

$$\|\hat{\zeta} - \zeta\|_\infty \leq C \sqrt{\frac{\ln p}{n}}. \tag{3.5}$$

Assumption 2 implies $\zeta_i \neq 0$ if and only if $\zeta_i > \mathcal{C}_0$ for some constant $\mathcal{C}_0 > 0$. By Theorem 2, if $\zeta_i = 0$ then $\hat{\zeta}_i = O((n^{-1} \ln p)^{1/2})$ with high probability. Let $\tilde{\zeta}$ be a hard thresholding of $\hat{\zeta}$, namely

$$\tilde{\zeta}_j = \hat{\zeta}_j \mathbb{1}(|\hat{\zeta}_j| > t_n), \qquad j \in [p] \tag{3.6}$$

where $t_n = (n^{-1} \ln p)^\alpha$ for some $0 < \alpha < \frac{1}{2}$. Given $\tilde{\zeta}$, define the active set $\tilde{S} = \{j : \tilde{\zeta}_j \neq 0\}$. The mis-classification rate for lda $\circ$ pca *conditional* on the training data $\{X_{11}, \ldots, X_{1n_1}\}$ and $\{X_{11}, \ldots, X_{1n_1}\}$, is then

$$\hat{R}_{\text{lda}\circ\text{pca}} = \mathbb{P}(\text{label}(\mathbf{Z}) \neq \hat{\Upsilon}_{\text{lda}\circ\text{pca}}(\mathbf{Z}) \mid \{X_{11}, \ldots, X_{1n_1}\}, \{X_{21}, \ldots, X_{2n_2}\}) \tag{3.7}$$

The following result shows that $\tilde{S}$ recovers $S_\zeta$ exactly and lda $\circ$ pca is asymptotically Bayes-optimal.

**Theorem 3.** *Suppose that $\mathbf{Z} \sim \pi_1 \mathcal{N}_p(\mu_1, \Sigma) + (1 - \pi_1)\mathcal{N}_p(\mu_2, \Sigma)$ where $\pi_1 \in (0, 1)$. Suppose Assumption 1 through Assumption 4 are satisfied. We then have*

$$\mathbb{P}(\tilde{S} \neq \mathcal{S}_\zeta) = \mathcal{O}(p^{-2}). \tag{3.8}$$

*Furthermore, suppose that $\ln p = o(n)$. We then have*

$$|\hat{R}_{\text{lda}\circ\text{pca}} - R_F| \longrightarrow 0 \tag{3.9}$$

*almost surely as $n, p \to \infty$.*

Our theoretical framework and results can be extended to elliptical distributions with minimal modifications; see the discussion in Section 7.3. Key remarks on our main results are presented below.

*Remark* 3. Theorem 2 and Theorem 3 appear, at first blush, quite similar to the results in Mai et al. (2012) for lassoed LDA using the discriminant direction $\beta = \Sigma^{-1}(\mu_1 - \mu_2)$. However the underlying assumptions behind these results are very different. In particular lassoed LDA relies on irrepresentable condition analogous to those in Zhao & Yu (2006) to achieve consistent variable selection, and thus the discriminative variables cannot be highly correlated with the remaining (irrelevant) variables. In contrast Theorem 3 do not assume an irrepresentable condition and furthermore Assumption 1 only assume boundedness of the non-spiked eigenvalues but allows for diverging spiked eigenvalues.

*Remark* 4. We now compare our results and settings with those in Sifaou et al. (2020a). Firstly, Sifaou et al. (2020a) assumed: (1) proportional growth rate $p/n \to c$ for some *finite* constant $c > 0$, (2) *distinct* spiked eigenvalues $\lambda_1 > \lambda_2 > \cdots > \lambda_d$ and lastly (3) $\|\Sigma\| = \mathcal{O}(1)$. In contrast, our work is derived under a more challenging set-up wherein (1) the dimension $p$ and the number of training samples $n$ satisfy $n^{-1} \ln p \to 0$ (Assumption 3), (2) some or even all of the eigenvalues $\lambda_1 \geq \cdots \geq \lambda_d > 0$ can be equal (Assumption 1) and (3) the leading eigenvalues $\{\lambda_k\}_{k=1}^d$ can diverge with $n$ (Assumption 3). We emphasize that $n^{-1} \ln p \to 0$ allows for $p = n^\gamma$ for any $\gamma > 1$, so that $p/n \to \infty$.

The assumptions used in Sifaou et al. (2020a) are because their theoretical analysis rely heavily on existing results in random matrix theory as presented in Donoho & Ghorbani (2018); Donoho et al. (2018), which generally require $p/n < \infty$ as well as bounded eigenvalues. In contrast, if $p/n \to \infty$, then Donoho & Ghorbani (2018) shows that the best estimate of the true covariance matrix is typically a diagonal matrix (see also Bickel & Levina (2004)), which might make the results in Sifaou et al. (2020a) sub-optimal for the high-dimensional regime $p/n \to \infty$ considered in our paper.

Finally, Sifaou et al. (2020a) *did not* guarantee that their classification rule is Bayes-optimal. Rather, they only show that the its mis-classification rate converge to some expression given in Theorem 3 of Sifaou et al. (2020a), and is strictly larger than Bayes error when $p/n$ increases. Sifaou et al. (2020b) subsequently extended the results in Sifaou et al. (2020a) to the case of heterogeneous covariance matrices but the resulting classifier is once again not Bayes-optimal. In contrast, Theorem 3 and Theorem 6 showed that the error rate for lda $\circ$ pca and qda $\circ$ pca (Quadratic Discriminant Analysis, QDA) converge to the Bayes error rate in the case of equal and unequal (class-conditional) covariance matrices, respectively. These results are, to the best of our knowledge, are among the first to show Bayes consistency when combining PCA with LDA/QDA for high-dimensional classification under divergent spikes. A key reason for this improvement

lies in two elements: (1) the assumption of sparsity in the signal vector $\zeta$, which mitigates estimation error high-dimensional means and covariances (motivated by Tibshirani et al. (2002); Fan & Fan (2008)) and (2) the use of the entrywise matrix perturbation bounds, providing precise control over the estimated eigenspace $\hat{\mathcal{U}}$ and the resulting whitened direction $\hat{\zeta}$.

## 4  Numerical results

We now present simulation results and real data analysis for lda ∘ pca. Comparisons will be made against the nearest shrunken centroids method (NSC) of Tibshirani et al. (2002), sparse linear discriminant analysis (SLDA) of Shao et al. (2011), direct sparse discriminant analysis (DSDA) of Mai et al. (2012), adaptive linear discriminant analysis (AdaLDA) of Cai & Zhang (2019), and LDA rule with CAT scores (CAT-LDA) of Zuber & Strimmer (2009); note that CAT-LDA uses a different sphering transformation compared to lda ∘ pca. There are three other commonly used classifiers that are not included in our comparisons, namely the naive Bayes rule, the linear programming discriminant (LPD) (Cai & Liu, 2011) and the regularized optimal affine discriminant (ROAD)(Fan et al., 2012). We omit these classifiers because (1) the naive Bayes rule is a special case of the NSC rule without soft thresholding of the mean vectors (2) Mai & Zou (2013) showed that the ROAD and DSDA classifiers are equivalent and (3) Cai & Zhang (2019) showed that the AdaLDA rule is a refinement of the LPD rule, i.e., compared to the LPD rule, the AdaLDA rule allows for "heteroscedastic constraints" and requires no tuning parameters.

Implementations of the DSDA and CAT-LDA rules are based on the TULIP and sda library in `R` while the implementation of the AdaLDA rule is based on `Matlab` codes provided in ADAM github repository. We also consider the rotation pre-processing step of Hao et al. (2015), which yields the transformed data $\{\mathcal{U}_{\mathrm{rot},n}^\top X_{11}, \ldots, \mathcal{U}_{\mathrm{rot},n}^\top X_{1n_1}\}$ and $\{\mathcal{U}_{\mathrm{rot},n}^\top X_{21}, \ldots, \mathcal{U}_{\mathrm{rot},n}^\top X_{2n_2}\}$ based on `Matlab` codes from HDRotation where we set $\gamma = 0.25$ in Eq. (2.10). These transformed data are then used as input to the DSDA and AdaLDA rules; we denote the resulting classifiers as DSDA(rot) and AdaLDA(rot).

### 4.1  Simulated examples

We consider three different simulation settings. For each setting, the number of features is set to $p = 800$ and we generate $n_1 = n_2 = 100$ data points from each class for the training data and also generate $n_1 = n_2 = 100$ data points from each class for the testing data. The classification accuracy of the classifiers in each simulation setting are computed based on 200 Monte Carlo replications. The mean vectors of the two classes are $\mu_1 = \mathbf{0}_{800}$ and $\mu_2 = (\mathbf{1}_{10}, \mathbf{0}_{790})$, i.e., the vector $\mu_2$ contains 10 non-zero entries with values all equal to 1. We consider the following models for the covariance matrix $\Sigma$. These models were considered previously in Fan et al. (2012); Mai et al. (2012); Cai & Zhang (2019); Cai & Liu (2011), among others. Throughout this sub-section, the number of selected features refers to the dimensionality of the representation on which each method operates. In particular, lda ∘ pca performs feature selection in the whitened feature space, whereas other methods select features directly from the raw feature space. See Section 7.2 for a detailed study in which the sparsity levels of the mean difference $\mu_2 - \mu_1$, the discriminant direction $\beta$, and the whitened direction $\zeta$ are matched.

1. *Model 1 (equal correlation)*: Here $\Sigma = (\sigma_{ij})_{p \times p} = \rho \mathbf{1}\mathbf{1}^\top + (1 - \rho)\mathcal{I}_p$, i.e., $\sigma_{ij} = \rho$ for $i \neq j$ and $\sigma_{ij} = 1$ for $i = j$. With this covariance structure $\Sigma$, the discriminant direction $\beta = \Sigma^{-1}(\mu_1 - \mu_0)$ and whitened direction $\zeta = \Sigma^{-1/2}(\mu_1 - \mu_0)$ are non-sparse (all entries of $\beta$ and $\zeta$ are non-zero).

2. *Model 2 (block diagonal with equal correlation)* Here $\Sigma = (\sigma_{ij})_{p \times p}$ is assumed to be a block diagonal matrix with two blocks of size $20 \times 20$ and $(p - 20) \times (p - 20)$. Both diagonal blocks are also of the form $\rho \mathbf{1}\mathbf{1}^\top + (1 - \rho)\mathcal{I}$ where the correlation $\rho$ is the same for both blocks. The discriminant direction $\beta$ and whitened direction $\zeta$ are sparse in this model, with $\beta$ and $\zeta$ both having 20 non-zero entries.

3. *Model 3 (random correlation)* Here $\Sigma = \mathcal{L}\mathcal{L}^\top + c_\mathcal{L}\mathcal{I}_p$ where $\mathcal{L} \in \mathbb{R}^{p \times 10}$ with $\mathcal{L}_{ij}$ generated from $\mathcal{N}(0, 1)$ and $c_\mathcal{L} = \min_{i \in [p]}[\mathcal{L}\mathcal{L}^\top]_{ii}$. Note that we generate a new $\mathcal{L}$ for every Monte Carlo replicate. For further comparison, we also consider $\mathcal{L}_{ij}$ generated from the uniform distribution on $[-1, 1]$

Table 1: Mis-classification rate (%) with standard deviations (%) in parentheses for equal correlation setting (model 1), based on 200 independent Monte Carlo replicates.

| $\rho$ | 0.50 | 0.60 | 0.70 | 0.80 | 0.90 |
|---|---|---|---|---|---|
| Oracle | 1.37 (0.87) | 0.61 (0.53) | 0.19 (0.30) | 0.19 (0.30) | 0.00 (0.00) |
| lda ∘ pca | 1.74 (1.00) | 1.00 (0.82) | 0.55 (0.67) | 0.55 (0.67) | 0.22 (0.39) |
| CAT-LDA | 7.18 (2.22) | 4.70 (1.77) | 2.60 (1.34) | 1.10 (0.88) | 0.38 (0.56) |
| DSDA | 3.27 (1.47) | 1.96 (1.08) | 0.74 (0.62) | 0.76 (0.65) | 0.00 (0.04) |
| DSDA(rot) | 6.31 (1.83) | 3.26 (1.27) | 1.16 (0.83) | 0.30 (0.41) | 0.01 (0.06) |
| AdaLDA | 4.15 (1.61) | 2.62 (1.24) | 1.22 (0.91) | 0.31 (0.42) | 0.00 (0.04) |
| AdaLDA(rot) | 7.04 (2.07) | 3.70 (1.26) | 1.19 (0.79) | 0.13 (0.26) | 0.00 (0.00) |
| SLDA | 17.97 (3.53) | 14.46 (2.99) | 11.14 (2.43) | 11.14 (2.43) | 1.64 (0.92) |
| NSC | 20.38 (8.53) | 22.60 (8.38) | 24.64 (8.36) | 24.64 (8.36) | 29.26 (8.04) |

and the Student's t-distribution with 5 degrees of freedom. The discriminant direction $\beta$ and $\zeta$ are generally non-sparse in this model, i.e., all of their entries are non-zero.

The tuning parameters for each classifier are chosen using five-fold cross validation (CV). In particular, SLDA (Shao et al., 2011) requires two tuning parameters, one being the number of non-zero entries in $\mu_1 - \mu_2$ and the other being the number of non-zero entries in $\Sigma$. For simplicity we shall assume that the sparsity of $\mu_1 - \mu_2$ is known when implementing SLDA and thus the only tuning parameter required is the number of non-zero entries in $\Sigma$; following Cai & Zhang (2019), this tuning parameter is selected from the set of values $\{\sqrt{n^{-1}\ln p}, 1.5\sqrt{n^{-1}\ln p}, \ldots, 5\sqrt{n^{-1}\ln p}\}$ using five-fold CV. The lda ∘ pca classifier in Algorithm 1 also requires two tuning parameters, namely (1) the number of spikes $d$ in the estimation of $\Sigma$ and (2) the sparsity level $s$ in $\zeta$. We chose $d$ to account for at least 90% of the total variability in the data, i.e., $d$ is the smallest integer of $k$ satisfying $(\sum_{i=1}^{k} \hat{\lambda}_i)/\mathbf{tr}(\hat{\Sigma}) \geq 0.9$; here $\hat{\lambda}_1 \geq \hat{\lambda}_2 \geq \ldots$ are the eigenvalues of the pooled sample covariance matrix $\hat{\Sigma}$. We acknowledge that the choice of $d$ can significantly impact performance; therefore, a sensitivity analysis is provided in Section 7.2. For each Monte Carlo replication, the training data are randomly split into five folds for CV, with new splits generated each time. The sparsity level $s$ is chosen from $s \in \{1, 2, \ldots, 30\}$ to minimize the average CV misclassification error. In the case where multiple values of $s$ yield the same minimum error, we choose the smallest such $s$ to promote a more parsimonious model. Similarly, the number of features in CAT-LDA is selected among the top $\{1, 2, \ldots, 30\}$ features ranked by CAT scores using five-fold CV. The choice of the upper limit 30 is motivated by the common sparsity condition $s_0 = \mathcal{O}((n/\ln p)^\tau)$, for some $\tau > 0$, for instance, $\tau = 1/2$ in Cai & Liu (2011). We deliberately choose $n/\ln p \, (\approx 30)$ as a relaxed upper bound to broaden the grid and ensure adequate coverage.

For data generated according to Model 1, Table 1 and Table 2 show that lda ∘ pca achieves the highest accuracy while using only a small number of features compared to the other classifiers. Note that although CAT-LDA also apply a whitening transformation before performing LDA, the accuracy of CAT-LDA is much worse compared to that of lda ∘ pca. The DSDA and DSDA(rot) classifiers have slightly better accuracies compared to those for the AdaLDA and AdaLDA(rot) classifiers; recall that the DSDA(rot) and AdaLDA(rot) classifiers first applied the rotation pre-processing step of Hao et al. (2015) before running DSDA and AdaLDA on the transformed data. The NSC classifier has the largest mis-classification error; this is a consequence of the NSC rule ignoring the correlation structure in $\Sigma$. Finally, the oracle classifier correspond to the LDA rule where $\pi_1, \mu_0, \mu_1$ and $\Sigma$ are known, and thus its error rate is the Bayes error rate from Eq. (3.1).

*Remark* 5. Recall that for Model 1, all entries of the discriminant direction $\beta$ and the whitened direction $\zeta$ are non-zero. These entries however can be classified into those representing strong signals vs weak signals based on their magnitudes; see Table 9 in the supplementary. The strong signals appear in the first 10 entries of $\beta$ (similarly $\zeta$) and the remaining $p - 10$ entries of $\beta$ (similarly $\zeta$) correspond to the weak signals. For example, if $\rho = 0.5$ then the first 10 entries of $\beta$ are all equal to 1.98 and the remaining $p = 10$ entries are all equal to $-0.02$. From Table 2 we see that lda ∘ pca kept all of the coordinates corresponding to the strong

Table 2: Average number of nonzero coefficients with standard deviations in parentheses for equal correlation setting (model 1), based on 200 independent Monte Carlo replicates.

| $\rho$ | 0.50 | 0.60 | 0.70 | 0.80 | 0.90 |
|---|---|---|---|---|---|
| lda ∘ pca | 12.04 (4.53) | 11.31 (4.10) | 9.52 (4.01) | 9.52 (4.01) | 3.68 (0.98) |
| CAT-LDA | 24.85 (3.74) | 24.70 (3.58) | 24.60 (3.97) | 22.29 (3.93) | 15.45 (2.39) |
| DSDA | 96.20 (31.06) | 106.42 (32.57) | 117.15 (30.52) | 117.05 (29.97) | 96.48 (8.66) |
| DSDA(rot) | 33.78 (32.71) | 36.88 (38.13) | 57.23 (61.02) | 147.57 (66.97) | 176.29 (5.08) |
| AdaLDA | 46.54 (5.95) | 45.84 (5.25) | 46.72 (6.41) | 47.86 (5.88) | 48.65 (5.02) |
| AdaLDA(rot) | 5.68 (1.82) | 5.57 (1.84) | 5.86 (1.82) | 7.83 (2.54) | 19.85 (9.96) |
| SLDA | 728.84 (226.63) | 788.16 (95.69) | 799.96 (0.21) | 799.96 (0.21) | 799.96 (0.21) |

signals and only added a few coordinates corresponding to the weak signals; this explains the small number of features used in lda ∘ pca. In contrast, the DSDA and SLDA classifiers include a large number of (noisy) features with weak signals.

For Model 2, Table 3 shows that SLDA performs slightly better than lda ∘ pca and DSDA. However, from Table 4, we see that SLDA also selects a much larger number of features compared to both lda∘pca and DSDA. Recall that, for model 2, both $\beta$ and $\zeta$ contains exactly 20 non-zero entries and hence SLDA is selecting a large number of extraneous, non-informative features; a similar, albeit much less severe, phenomenon is observed for DSDA. Table 3 and Table 4 shows that lda∘pca, CAT-LDA, and AdaLDA have comparable accuracy with a similar number of selected features. The NSC rule once again has the largest mis-classification error due to it ignoring the correlation structure in $\Sigma$. Finally we see that the rotation pre-processing step described in Hao et al. (2015) lead to a substantial loss in accuracy for the DSDA and AdaLDA classifiers; to understand why this happens, we need to extend the theoretical analysis in Hao et al. (2015) (which assume that $\Sigma$ and $\mathcal{U}_{\text{rot},p}$ are known) to the setting where $\Sigma$ and $\mathcal{U}_{\text{rot},p}$ have to be estimated. We leave this investigation for future work.

Finally for Model 3, Table 5 and Table 6 show that lda ∘ pca has both the highest accuracy as well as the smallest number of selected features among all the considered classifiers; DSDA has a slightly worse accuracy and also selected a much larger number of features, when compared to lda ∘ pca. SLDA now has the worst accuracy and also selects almost all $p = 800$ features, and this is a consequence of $\Sigma$ being a dense matrix. The mis-classification rate for AdaLDA and AdaLDA(rot) are also quite large, and we surmise that this is due to the numerical instability when solving the linear programming problem in AdaLDA. Indeed, the condition numbers for $\Sigma$ can be quite large; see Table 12 in the supplementary for summary statistics of these condition numbers using the same 200 Monte Carlo replicates as that for generating Table 5. Following the suggestion in Cai & Zhang (2019), we replace the sample covariance matrix $\hat{\Sigma}$ used in the optimization problem for AdaLDA and AdaLDA(rot) (see Eq. (8) and Eq. (9) in Cai & Zhang (2019)) with $\tilde{\Sigma} = \hat{\Sigma} + \sqrt{n^{-1}\ln p}\,\mathcal{I}_p$; the resulting classifiers are denoted as AdaLDA(reg) and AdaLDA(rot + reg), respectively. Table 5 however shows that using $\tilde{\Sigma}$ only leads to a minimal increase in accuracy. Finally we observe that the rotation pre-processing step once again leads to a substantial loss in accuracy for the DSDA and AdaLDA classifiers.

*Remark* 6. While our theoretical results primarily focus on fixed sparsity (see Remark 1 for more on growing sparsity), we deliberately include dense examples with $s_0 = p$, such as Model 1 and Model 3, to demonstrate robustness beyond the sparse regime. In the dense models, full recovery of $\zeta$ or $\beta$ is not anticipated but lda ∘ pca still delivers strong classification performance due to effective decorrelation. In particular, for Model 1, our method's performance is very close to the oracle across varying $\rho$.

## 4.2 Real data analysis

We now assess the performance of lda ∘ pca on two gene expression data sets for leukaemia and lung cancer. The leukaemia dataset (Golub et al., 1999) includes $p = 7128$ gene measurements on 72 patients with either acute lymphoblastic leukemia (ALL) or acute myeloid leukemia (AML) and we wish to classify patients into

Table 3: Mis-classification rate (%) with standard deviations (%) in parentheses for block diagonal setting (model 2), based on 200 independent Monte Carlo replicates.

| $\rho$ | 0.50 | 0.60 | 0.70 | 0.80 | 0.90 |
|---|---|---|---|---|---|
| Oracle | 5.30 (1.71) | 5.21 (1.52) | 5.34 (1.64) | 5.48 (1.62) | 5.40 (1.74) |
| lda ∘ pca | 9.08 (2.71) | 8.61 (2.59) | 8.30 (2.55) | 8.26 (2.45) | 8.12 (2.58) |
| CAT-LDA | 10.08 (3.00) | 9.36 (2.57) | 9.31 (2.50) | 9.25 (2.59) | 9.21 (2.89) |
| DSDA | 9.63 (2.61) | 9.62 (2.64) | 9.50 (2.66) | 9.58 (2.67) | 9.47 (2.42) |
| DSDA(rot) | 20.64 (4.16) | 20.52 (3.99) | 20.75 (4.18) | 21.02 (4.44) | 20.90 (4.40) |
| AdaLDA | 12.70 (2.99) | 13.06 (3.39) | 13.88 (3.45) | 13.75 (3.48) | 14.68 (3.86) |
| AdaLDA(rot) | 23.91 (3.79) | 23.81 (3.72) | 24.33 (4.10) | 23.89 (3.92) | 24.12 (3.94) |
| SLDA | 6.99 (2.89) | 6.66 (1.92) | 6.71 (2.17) | 6.62 (1.97) | 6.56 (1.90) |
| NSC | 25.14 (3.11) | 26.47 (3.31) | 28.61 (3.35) | 29.59 (3.24) | 30.52 (3.82) |

Table 4: Average number of nonzero coefficients with standard deviations in parentheses for block diagonal setting (model 2), based on 200 independent Monte Carlo replicates.

| $\rho$ | 0.50 | 0.60 | 0.70 | 0.80 | 0.90 |
|---|---|---|---|---|---|
| lda ∘ pca | 20.19 (4.87) | 20.48 (5.06) | 20.48 (4.88) | 20.96 (4.96) | 20.58 (4.44) |
| CAT-LDA | 20.02 (4.95) | 19.86 (4.31) | 20.05 (4.62) | 20.32 (4.21) | 20.33 (4.54) |
| DSDA | 50.28 (26.61) | 50.74 (28.92) | 47.48 (25.77) | 43.66 (22.42) | 45.15 (23.78) |
| DSDA(rot) | 51.02 (39.23) | 46.13 (29.09) | 51.93 (36.08) | 45.56 (31.15) | 49.62 (33.50) |
| AdaLDA | 19.45 (2.37) | 18.53 (2.23) | 17.93 (2.27) | 17.40 (2.20) | 16.85 (2.07) |
| AdaLDA(rot) | 14.40 (3.14) | 14.81 (3.53) | 15.16 (3.40) | 15.69 (3.99) | 14.97 (3.84) |
| SLDA | 405.98 (390.84) | 320.17 (380.33) | 210.99 (336.12) | 242.19 (352.81) | 172.00 (309.62) |

either ALL or AML based on their gene expressions. The training set consists of the gene expressions for $n_1 = 27$ patients with ALL and $n_2 = 11$ patients with AML while the test set contains the gene expression data for 20 patients with ALL and 14 patients with AML. The lung cancer data was originally analyzed in Gordon et al. (2002) and we use a version of the data wherein the predictor variables with low variances are removed; see Pun & Hadimaja (2021). The resulting data set contains tumor tissues with $p = 1577$ features collected from patients with adenocarcinoma (AD) or malignant pleural mesothelioma (MPM). According to Gordon et al. (2002), MPM is highly lethal but distinguishing between MPM and AD is quite challenging in both clinical and pathological settings. The training set consists of $n_1 = 120$ gene expressions for patients with AD and $n_2 = 25$ gene expressions for patients with MPM, and the test set consists of gene expressions for 30 patients with AD and 6 patients with MPM. Table 7 and Table 8 presents the classification accuracy and number of selected features for various classifiers when applied to the leukemia and lung cancer data, respectively. The hyperparameters for these classifiers are selected using leave-one-out CV. Table 7 and 8 indicate that the performance of lda ∘ pca is competitive with existing state-of-the-art classifiers while also operating on a substantially lower-dimensional representation.

## 5 Extensions of lda ∘ pca

### 5.1 Multi-class classification

Suppose we are given training data from $K \geq 3$ classes where the feature vectors for each class are iid samples from a $p$-dimensional multivariate normal distribution, i.e.,

$$X_{i1}, \ldots, X_{in_i} \sim \mathcal{N}_p(\mu_i, \Sigma), \qquad i \in [K].$$

Table 5: Mis-classification rate (%) with standard deviations (%) in parentheses for random correlation setting (model 3), based on 200 independent Monte Carlo replicates.

| Method | $\mathcal{U}(-1,1)$ | $\mathcal{N}(0,1)$ | $\mathbf{t}_5$ |
|---|---|---|---|
| lda ∘ pca | 5.07 (2.40) | 12.39 (4.17) | 13.72 (5.00) |
| CAT-LDA | 11.93 (4.04) | 23.67 (6.13) | 25.68 (6.82) |
| DSDA | 7.32 (3.05) | 16.92 (5.21) | 17.90 (5.49) |
| DSDA(rot) | 18.51 (6.37) | 31.81 (7.11) | 33.65 (7.28) |
| AdaLDA | 17.94 (6.03) | 44.55 (7.12) | 47.89 (5.10) |
| AdaLDA(rot) | 23.00 (7.49) | 39.29 (6.11) | 42.52 (6.28) |
| AdaLDA(reg) | 29.48 (7.58) | 47.49 (4.52) | 48.96 (4.04) |
| AdaLDA(reg+rot) | 21.23 (7.43) | 38.93 (6.60) | 42.52 (6.28) |
| SLDA | 50.10 (4.40) | 49.98 (4.25) | 50.09 (3.70) |
| NSC | 30.30 (6.28) | 44.55 (5.16) | 46.70 (4.49) |

Table 6: Model size with standard deviations in parentheses for random correlation setting (model 3), based on 200 independent Monte Carlo replicates.

| Method | $\mathcal{U}(-1,1)$ | $\mathcal{N}(0,1)$ | $\mathbf{t}_5$ |
|---|---|---|---|
| lda ∘ pca | 11.93 (4.10) | 11.48 (3.88) | 11.37 (3.92) |
| CAT-LDA | 21.19 (6.51) | 20.64 (6.88) | 20.75 (6.90) |
| DSDA | 66.00 (31.92) | 65.63 (34.45) | 64.91 (30.51) |
| DSDA(rot) | 51.08 (34.82) | 71.94 (41.64) | 77.56 (49.29) |
| AdaLDA | 15.99 (2.32) | 10.46 (1.97) | 8.51 (2.22) |
| AdaLDA(rot) | 14.62 (4.61) | 19.08 (3.93) | 16.26 (3.43) |
| AdaLDA(reg) | 11.38 (1.77) | 8.50 (2.00) | 7.12 (1.96) |
| AdaLDA(reg+rot) | 9.86 (4.71) | 15.99 (3.63) | 16.26 (3.43) |
| SLDA | 799.51 (1.31) | 800.00 (0.07) | 800.00 (0.00) |
| NSC | 12.21 (8.47) | 55.38 (131.65) | 102.89 (192.83) |

Here $n_i$ denote the number of training data points from class $i \in [K]$. The testing data point $\mathbf{Z}$ is drawn from a mixture of $K$ multivariate normal distributions namely, $\mathbf{Z} \sim \sum_{i=1}^{K} \pi_i \mathcal{N}_p(\mu_i, \Sigma)$ with $\pi_i \geq 0$ and $\sum_{i=1}^{K} \pi_i = 1$. Note that, for ease of exposition, the $\{X_{ij}\}$ are assumed to be multivariate normals but the subsequent results also hold when $\{X_{ij}\}$ are elliptically distributed as in Section 7.3.

Now define, for $2 \leq i \leq K$, the whitened direction $\zeta^{(i)}$ and the whitened indices $\mathcal{S}_i$ via

$$\zeta^{(i)} = \mathcal{W}(\mu_i - \mu_1), \qquad \mathcal{S}_i = \{j : \zeta_j^{(i)} \neq 0\}.$$

Define the global whitened set as $\mathcal{S}_\zeta = \mathcal{S}_2 \cup \mathcal{S}_3 \cup \cdots \cup \mathcal{S}_K$. The indices in $\mathcal{S}_\zeta$ are the important variables for feature selection. The extension of lda ∘ pca to $K \geq 3$ classes is then given in Algorithm 3 below, and theoretical results are presented in the supplementary.

Table 7: Mis-classification rate and model size of various methods for the leukaemia data

| Method | Training Error | Test Error | Model Size |
|---|---|---|---|
| lda ∘ pca | 0/38 | 1/34 | 12 |
| DSDA | 0/38 | 1/34 | 36 |
| AdaLDA (reg) | 0/38 | 1/34 | 18 |
| NSC | 1/38 | 3/34 | 24 |

Table 8: Mis-classification rate and model size of various methods for the lung cancer data

| Method | Training Error | Test Error | Model Size |
|---|---|---|---|
| lda ∘ pca | 0/145 | 0/36 | 20 |
| DSDA | 1/145 | 0/36 | 25 |
| AdaLDA (reg) | 0/145 | 0/36 | 388 |
| NSC | 1/145 | 0/36 | 1206 |

---

**Algorithm 3:** $K$-classes lda ∘ pca decision rule

---

**Input:** $\bar{X}_i,\ i \in [K]$, $\hat{\Sigma}$ and the test sample $\mathbf{Z}$
**Output:** $\hat{\Upsilon}_{\mathrm{lda \circ pca}}(\mathbf{Z})$
**Algorithm**

// Step 1: Perform PCA on the feature vectors for the training data (standard PCA approach). Extract the $d$ largest principal components to obtain $\hat{\mathcal{W}}$.

// Step 2: For $2 \le i \le K$, let $\tilde{X}_i = \hat{\mathcal{W}}\bar{X}_i$ and $\hat{\zeta}^{(i)} = \tilde{X}_i - \tilde{X}_1$. Also let $\hat{\mathcal{S}}_i$ be the set of indices corresponding to the $s_i$ largest elements of $\hat{\zeta}^{(i)}$ in absolute values. The value of $s_i$ is, in general, a user-specified or tuning parameter. Nevertheless, under certain conditions, we can also estimate $s_i$; see Eq. (8.51) in the supplementary.

// Step 3: Given the test data point $\mathbf{Z}$, let $\tilde{\mathbf{Z}} = \hat{\mathcal{W}}\mathbf{Z}$ and denote by $\hat{\zeta}^{(i)}_{\hat{\mathcal{S}}_i}$ the vector obtained from $\hat{\zeta}^{(i)}$ by keeping only those coordinates belonging to $\hat{\mathcal{S}}_i$.

// Step 4: Set $\tilde{D}_1 = 0$ and calculate, for $2 \le i \le K$, the discriminant score for class $i$ relative to class 1 as

$$\tilde{D}_i = \big[\tilde{\mathbf{Z}} - \big(\tfrac{\tilde{X}_i + \tilde{X}_1}{2}\big)\big]^\top_{\hat{\mathcal{S}}_i} \hat{\zeta}^{(i)}_{\hat{\mathcal{S}}_i} + \ln \frac{n_i}{n_1}. \tag{5.1}$$

.

// Step 5: Assign the label of $\mathbf{Z}$ to the class that maximizes the discriminant score, i.e.,

$$\hat{\Upsilon}_{\mathrm{lda \circ pca}}(\mathbf{Z}) = \arg\max_{i \in [K]} \tilde{D}_i. \tag{5.2}$$

---

## 5.2 Heterogeneous covariance matrices

Despite the simplicity and popularity of regularized or sparse LDA for high-dimensional data, the assumption of equal covariances is not always tenable in practice. More specifically, suppose we are given a $p$-variate random vector $\mathbf{Z}$ drawn from a mixture $\pi_1 \mathcal{N}_p(\mu_1, \Sigma_1) + (1 - \pi_1)\mathcal{N}_p(\mu_2, \Sigma_2)$ with $\Sigma_1$ possibly distinct from

$\Sigma_2$. The Bayes classifier is then the QDA rule given by

$$\Upsilon_F(\mathbf{Z}) = \begin{cases} 1 & \text{if } (\mathbf{Z} - \mu_1)^\top \Sigma_1^{-1} (\mathbf{Z} - \mu_1) - (\mathbf{Z} - \mu_2)^\top \Sigma_2^{-1} (\mathbf{Z} - \mu_2) \leq \kappa \\ 2 & \text{otherwise} \end{cases} \tag{5.3}$$

where $\kappa = 2 \ln \frac{\pi_1}{1 - \pi_1} - \ln \frac{|\Sigma_1|}{|\Sigma_2|}$. If $\Sigma_1 = \Sigma_2$, then Eq. (5.3) reduces to Eq. (1.1). We now discuss how the results in Section 3 can be extended to quadratic discriminant analysis (QDA) with PCA. Firstly we assume that $\Sigma_1$ and $\Sigma_2$ both have spiked covariance structures as specified below.

*Assumption* 5. Let $\boldsymbol{u}_{i1}, \ldots, \boldsymbol{u}_{id_i}$, for $i = 1, 2$ be orthonormal vectors in $\mathbb{R}^p$ and assume that the covariance matrix $\Sigma_i$ for the $p$-variate normal distributions $\mathcal{N}_p(\mu_i, \Sigma_i)$ is of the form

$$\Sigma_i = \sum_{k=1}^{d_i} \lambda_{ik} \mathbf{u}_{ik} \mathbf{u}_{ik}^\top + \sigma_i^2 \mathcal{I}_p = \mathcal{U}_i \Lambda_i \mathcal{U}_i^\top + \sigma_i^2 \mathcal{I}_p, \quad i = 1, 2. \tag{5.4}$$

Here $\Lambda_i = \mathrm{diag}(\lambda_{ik})$ is a $d_i \times d_i$ diagonal matrix and $\mathcal{U}_i$ is a $p \times d_i$ matrix with orthonormal columns. We assume implicitly that $\lambda_{i1} \geq \cdots \geq \lambda_{id} > 0$, $\sigma_i > 0$ and $d_i \ll p$, $i = 1, 2$.

Note that a recent line of research on QDA for high-dimensional classification is based on the assumption that $\Sigma_2^{-1} - \Sigma_1^{-1}$ is sparse; see e.g., Li & Shao (2015); Jiang et al. (2018); Cai & Zhang (2021). In contrast, Assumption 5 do not enforce any sparsity assumption and also allows for $\Sigma_1$ and $\Sigma_2$ to have different *spiked* eigenvalues and eigenvectors; the latter is is a generalization of the common principal components (CPC) assumption in Zhu (2006); Pepler et al. (2017); Flury (1988) where the leading principal components are the same for both $\Sigma_1$ and $\Sigma_2$.

Under Assumption 5, the whitening transformation $\mathcal{W}_i = \Sigma_i^{-1/2}$ for $i = 1, 2$ is of the form in Eq. (2.7) and thus a suitable estimate for $\mathcal{W}_i$ is given by Eq. (3.3). Let $\zeta_i = \mathcal{W}_i \mu_i$ for $i = 1, 2$, and denote the whitened index sets by $\mathcal{A}_i = \{j : \zeta_{ij} \neq 0\}$ for $i = 1, 2$. Let $\mathcal{A}_0 = \mathcal{A}_1 \cup \mathcal{A}_2$ and note that the elements in $\mathcal{A}_0$ are the *signal* coordinates for the QDA rule (after the PCA step). Eq. (5.3) can be written as

$$\Upsilon(\mathbf{Z}) := \begin{cases} 1 & \text{if } \|[\mathcal{W}_1(\mathbf{Z} - \mu_1)]_{\mathcal{A}_0}\|^2 - \|[\mathcal{W}_2(\mathbf{Z} - \mu_2)]_{\mathcal{A}_0}\|^2 \leq \kappa \\ 2 & \text{otherwise} \end{cases} \tag{5.5}$$

A plugin decision rule is then obtained by replacing $\mu_i, \Sigma_i$ and $\mathcal{A}_0$ with their estimates $\bar{X}_i, \hat{\Sigma}_i$ and $\hat{\mathcal{A}}_0$. Note that the intercept $\kappa$ is non-trivial to estimate in the high-dimensional setting as it involves the log-determinant $\ln \frac{|\Sigma_1|}{|\Sigma_2|}$; see Cai et al. (2015) for further details. For our paper we employ the data-driven approach of Jiang et al. (2018) which circumvents the need to estimate the determinants of $\Sigma_1$ and $\Sigma_2$. The full details of qda $\circ$ pca are specified in Algorithm 4 below, and theoretical results for qda $\circ$ pca are provided in the supplementary.

---

**Algorithm 4:** qda ∘ pca decision rule

---

**Input:** $\bar{X}_1$, $\bar{X}_2$, $\hat{\Sigma}_1$, $\hat{\Sigma}_2$ and a test data point $\mathbf{Z}$

**Output:** $\hat{\Upsilon}_{\text{qda∘pca}}(\mathbf{Z})$

**Algorithm**

> `// Step 1:` Perform PCA on the feature vectors for the training data (standard PCA approach). Extract, for $i = 1, 2$, the $d_i$ largest principal components $\hat{\mathcal{U}}_i$ and compute $\hat{\mathcal{W}}_i$ as in Eq. (3.3).
>
> `// Step 2:` For $i = 1, 2,$, set $\hat{\zeta}_i = \hat{\mathcal{W}}_i \bar{X}_i$ and form the indices set $\hat{\mathcal{A}}_i$ by selecting the $s_i$ largest (in magnitude) coordinates of $\hat{\zeta}_i$.
>
> `// Step 3:` Let $\hat{\mathcal{A}}_0 = \hat{\mathcal{A}}_1 \cup \hat{\mathcal{A}}_2$ and define for any $\boldsymbol{x} \in \mathbb{R}^p$
>
> $$\mathbf{Q}(x \mid \{\bar{X}_1, \bar{X}_2, \hat{\mathcal{W}}_1, \hat{\mathcal{W}}_2, \hat{\mathcal{A}}_0\}) = \left\|[\hat{\mathcal{W}}_1(\boldsymbol{x} - \bar{X}_1)]_{\hat{\mathcal{A}}_0}\right\|^2 - \left\|[\hat{\mathcal{W}}_2(\boldsymbol{x} - \bar{X}_2)]_{\hat{\mathcal{A}}_0}\right\|^2.$$
>
> .
>
> `// Step 4:` Find $\hat{\kappa}$ to minimize the empirical 0-1 loss of the decision rule induced by $\mathbf{Q}$, i.e.,
>
> $$\hat{\kappa} = \arg\min_{\eta \in \mathbb{R}} \frac{1}{n} \sum_{i=1}^{2} \sum_{j=1}^{n_i} \mathbb{1}(\Upsilon(X_{ij} \mid \{\bar{X}_1, \bar{X}_2, \hat{\mathcal{W}}_1, \hat{\mathcal{W}}_2, \hat{\mathcal{A}}_0, \eta_{\text{thresh}}\}) \neq i) \tag{5.6}$$
>
> where, for any $\eta_{\text{thresh}} \in \mathbb{R}$, we define
>
> $$\Upsilon(X_{ij} \mid \{\bar{X}_1, \bar{X}_2, \hat{\mathcal{W}}_1, \hat{\mathcal{W}}_2, \hat{\mathcal{A}}_0, \eta_{\text{thresh}}\}) = \begin{cases} 1 & \text{if } \mathbf{Q}(X_{ij} \mid \{\bar{X}_1, \bar{X}_2, \hat{\mathcal{W}}_1, \hat{\mathcal{W}}_2, \hat{\mathcal{A}}_0\}) \leq \eta_{\text{thresh}} \\ 2 & \text{otherwise} \end{cases}$$
>
> .
>
> `// Step 5:` Given a test data point $\mathbf{Z}$, return the decision rule
>
> $$\hat{\Upsilon}_{\text{qda∘pca}}(\mathbf{Z}) = \begin{cases} 1 & \text{if } \mathbf{Q}(\mathbf{Z} \mid \{\bar{X}_1, \bar{X}_2, \hat{\mathcal{W}}_1, \hat{\mathcal{W}}_2, \hat{\mathcal{A}}_0\}) \leq \hat{\kappa} \\ 2 & \text{otherwise} \end{cases} \tag{5.7}$$

---

As indicated by Jiang et al. (2018), $\eta_{\text{thresh}}$ is selected by minimizing the in-sample misclassification error based on $\bar{X}1$, $\bar{X}2$, $\hat{\Sigma}1$, and $\hat{\Sigma}2$. Under Assumption 5, the grid search can be narrowed to a neighborhood around $-2\log\left(\frac{|\hat{\mathcal{W}}2|}{|\hat{\mathcal{W}}1|}\right) - 2\log\left(\frac{n_1}{n_2}\right)$. This is justified by the improved stability of the log-determinant ratio in the spiked setting, where bulk eigenvalue $\sigma_i^2$ is consistently estimated via the pooled sample covariance and the leading spikes remain well-separated (from the bulks).

## 6    Discussion

In this paper we addressed the classification problem for high-dimensional data by analyzing the prototypical lda ∘ pca classifier that first transforms the feature vectors using a whitening transformation, then performs feature selection on the whitened data, and finally applies LDA in the dimensionally reduced space. We show that, under a spiked eigenvalue structure for $\Sigma$, the mis-classification error rate for lda∘pca is asymptotically Bayes optimal whenever $n \to \infty$ and $n^{-1}\ln p \to 0$. While the Bayes consistency of lda∘pca is similar to that of classifiers based on estimating the discriminant direction $\beta$, the underlying assumptions and motivations for our results are substantially different. Indeed, the focus on PCA and the whitening matrix leads to the natural assumption that $\Sigma$ has spiked/diverging eigenvalues while earlier results that focus on estimation of $\beta$ had generally assumed that $\Sigma$ is sparse or that the eigenvalues of $\Sigma$ are bounded. Numerical experiments indicate that lda∘pca is competitive with existing state-of-the-art classifiers while operating on a substantially

lower-dimensional representation. This behavior persists even when the underlying sparsity levels in the raw and whitened feature spaces are matched; see more details in Section 7.2.

We now mention two interesting issues for future research. The first is to extend the theoretical results in this paper for combining LDA with PCA to other, possibly non-linear, dimension reduction techniques such as (classical) multidimensional scaling, kernel PCA, and Laplacian eigenmaps (Belkin & Niyogi, 2003), followed by learning a classifier in the dimensionally reduced space. The second issue concerns the spiked covariance structure in Assumption 1. In particular, while Assumption 1 is widely used in the literature, see e.g., Hao et al. (2015); Cai et al. (2013); Johnstone (2001); Birnbaum et al. (2012), its assumption on the non-spiked eigenvalues and eigenvectors might be somewhat restrictive. We can consider relaxing Assumption 1 by assuming that $\Sigma$ arises from an approximate factor model (Fan et al., 2013) or that $\Sigma$ can be decomposed into a low-rank plus sparse matrix structure (Agarwal et al., 2011). We surmise that, due to the focus on the whitening matrix $\Sigma^{-1/2}$, theoretical analysis of lda $\circ$ pca under these more general covariance structure will also leads to interesting technical developments; e.g., while perturbation results for $\hat{\Sigma}^{-1} - \Sigma^{-1}$ given $\hat{\Sigma} - \Sigma$ are well-studied, much less is known about perturbation bounds for $\hat{\Sigma}^{-1/2} - \Sigma^{-1/2}$ given $\hat{\Sigma} - \Sigma$.

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

# 7  Supplementary Numerical Results

## 7.1  Supplementary Results to Model 1 to Model 3

We provide here several tables that supplement the simulation results in Section 4. Table 9 reports the magnitudes for the strong and weak signals in the discriminant direction $\beta$ and whitened direction $\zeta$ for Model 1. Table 10 and 11 reports the average number of strong signals and weak signals captured by lda ∘ pca, DSDA, AdaLDA and SLDA under Model 1. From Table 11 we see that lda ∘ pca includes only a few features corresponding to the weak signals and thus selects only a small number of features compared to the other classifiers. We also report in Table 12 the summary statistics for the condition numbers of the covariance matrix $\Sigma$ in Model 3; these statistics indicate that the performance of AdaLDA for Model 3 can be sub-optimal as AdaLDA estimates $\beta$ by solving a linear programming problem which is numerically unstable when $\hat{\Sigma}$ have large condition numbers.

Table 9: Strong and faint signals for equal correlation setting (model 1). See the discussion in Remark 5

| $\rho$ | 0.50 | 0.60 | 0.70 | 0.80 | 0.90 |
|---|---|---|---|---|---|
| strong signal $\beta$ | 1.98 | 2.47 | 3.29 | 4.94 | 9.88 |
| strong signal $\zeta$ | 1.40 | 1.56 | 1.80 | 2.21 | 3.12 |
| faint signal $\beta$ | -0.02 | -0.03 | -0.04 | -0.06 | -0.12 |
| faint signal $\zeta$ | -0.02 | -0.02 | -0.02 | -0.03 | -0.04 |

Table 10: Average number of strong discriminative and whitened variable with standard deviations in parentheses for equal correlation setting (model 1)

| $\rho$ | 0.50 | 0.60 | 0.70 | 0.80 | 0.90 |
|---|---|---|---|---|---|
| lda ∘ pca | 9.59 (0.82) | 9.42 (0.85) | 8.53 (1.41) | 8.53 (1.41) | 3.68 (0.98) |
| DSDA | 10.00 (0.00) | 9.99 (0.07) | 9.99 (0.07) | 9.99 (0.07) | 10.00 (0.00) |
| AdaLDA | 9.95 (0.22) | 9.93 (0.26) | 9.95 (0.23) | 10.00 (0.07) | 10.00 (0.07) |
| SLDA | 9.98 (0.22) | 9.98 (0.28) | 10.00 (0.00) | 10.00 (0.00) | 10.00 (0.00) |

Table 11: Average number of weak discriminative and whitened variable with standard deviations in parentheses for equal correlation setting (model 1)

| $\rho$ | 0.50 | 0.60 | 0.70 | 0.80 | 0.90 |
|---|---|---|---|---|---|
| lda ∘ pca | 2.44 (4.22) | 1.90 (3.73) | 0.98 (3.35) | 0.98 (3.35) | 0.00 (0.00) |
| DSDA | 86.20 (31.06) | 96.42 (32.57) | 107.16 (30.53) | 107.06 (29.96) | 86.48 (8.66) |
| ADaLDA | 36.59 (5.94) | 35.91 (5.22) | 36.78 (6.36) | 37.86 (5.88) | 38.65 (5.02) |
| SLDA | 718.86 (226.57) | 778.18 (95.53) | 789.96 (0.21) | 789.96 (0.21) | 789.96 (0.21) |

Table 12: Summary statistics of condition numbers of generated $\Sigma$ from uniform, normal and student distributions

| Statistics | $\mathcal{U}(-1,1)$ | $\mathcal{N}(0,1)$ | $\mathbf{t}_5$ |
|---|---|---|---|
| Mean | 415.3 | 688.0 | 1059.2 |
| SD | 102.4 | 195.2 | 400.0 |
| Median | 391.5 | 658.5 | 966.3 |
| IQR | 108.0 | 205.9 | 314.1 |
| Max | 849.9 | 1622.1 | 3708.4 |
| Min | 266.3 | 396.9 | 536.0 |

## 7.2 Sensitive Analysis

This subsection aims to explore the sensitivity of lda ∘ pca to the number of spikes $d$. As a baseline, we list the performance of Oracle and DSDA. Consider a covariance matrix $\Sigma = (\sigma_{ij})_{p \times p}$ with parameters $\rho_1, \rho_2 > 0$ and $\rho_1 + \rho_2 < 2$:

$$\Sigma = \rho_1 p \, \mathbf{u}_1 \mathbf{u}_1^\top + \rho_2 p \, \mathbf{u}_2 \mathbf{u}_2^\top + (2 - \rho_1 - \rho_2)(\mathcal{I}_p - \mathbf{u}_1 \mathbf{u}_1^\top - \mathbf{u}_2 \mathbf{u}_2^\top) \tag{7.1}$$

where $\mathbf{u}_1 = \frac{1}{\sqrt{p}} \mathbf{1}_p$, and $\mathbf{u}_2 = \frac{1}{\sqrt{2}}(\mathbf{e}_1 - \mathbf{e}_2)$.

This covariance structure is a slight modification of Model 1 (the equal correlation model) to incorporate two spikes corresponding to the eigenvalues $\rho_1 p$ and $\rho_2 p$. The class means are set to be $\mu_1 = \mathbf{0}_p$ and $\mu_2 = \mathbf{e}_1 + \mathbf{e}_2 - 2\mathbf{e}_3$ so that the mean difference $\mu_2 - \mu_1$, discriminant direction $\beta$ and whitened direction $\zeta$ are all sparse with exactly three nonzero entries, located at the first three coordinates. This contrasts with the dense structures exhibited in Model 1 and Model 3.

For sensitivity analysis, our method variants are labeled as lda ∘ pca($d$) for $d = 1, 2, 3$, with $d = 2$ being the correctly specified case. Training and test samples are generated using $\Sigma$ from Eq. (7.1), following Monte Carlo setups and CV procedure detailed in Section 4.1. Table 13 shows that correct specifying or overestimating $d$ results in classification errors comparable to DSDA whereas underestimating $d$ leads to a notable degradation in performance. Since overestimation does not harm accuracy, we recommend, in practice, selecting $d$ as the smallest integer such that the cumulative eigenvalue ratio $(\sum_{i=1}^d \hat{\lambda}_i)/\mathbf{tr}(\hat{\Sigma}) \geq 0.9$, ensuring 90% of the total variance is retained in the selected subspace.

DSDA achieves similar accuracy but tends to over-select non-informative features is as shown in Table 14 with average 49.46 selected features well above the true sparsity level of 3. In comparison, lda ∘ pca maintains comparable accuracy across all simulation settings. Our method also yields more parsimonious and offers greater computational efficiency especially under the spike covariance structure.

Table 13: Mis-classification rate (%) with standard deviations (%) in parentheses for Eq. (7.1), based on 200 independent Monte Carlo replicates.

| $(\rho_1, \rho_2)$ | (1.5, 0.3) |
|---|---|
| Oracle | 0.30 (0.36) |
| lda ∘ pca($d = 1$) | 24.19 (23.09) |
| lda ∘ pca($d = 2$) | 0.55 (0.60) |
| lda ∘ pca($d = 3$) | 0.55 (0.60) |
| DSDA | 0.50 (0.49) |

## 7.3 Elliptical distributions

A random vector $X \in \mathbb{R}^p$ is said to have an elliptical distribution with mean $\mu$ and covariance matrix $\Sigma$ if the probability density function for $X$ is of the form

$$f(\boldsymbol{x}) = \mathcal{C}_f |\Sigma|^{-1/2} h((\boldsymbol{x} - \mu)^\top \Sigma^{-1} (\boldsymbol{x} - \mu)), \quad \text{for all } \boldsymbol{x} \in \mathbb{R}^p. \tag{7.2}$$

Table 14: Average number of nonzero coefficients with standard deviations in parentheses for for Eq. (7.1), based on 200 independent Monte Carlo replicates.

| $(\rho_1, \rho_2)$ | (1.5, 0.3) |
|---|---|
| lda ∘ pca($d = 1$) | 1.49 (1.90) |
| lda ∘ pca($d = 2$) | 3.82 (4.28) |
| lda ∘ pca($d = 3$) | 3.71 (3.92) |
| DSDA | 49.35 (32.97) |

Here $\mathcal{C}_f$ is a normalization constant, $|\Sigma|$ is the determinant of $\Sigma$ and $h$ is a monotone function on $[0, \infty)$. The class of elliptical distributions is analytically tractable and many results that hold for multivariate normal distributions can be extended to general elliptical distributions. In particular, Fang & Anderson (1992) showed that Fisher's LDA rule in Eq. (1.1) is also Bayes optimal whenever the feature vectors $X$ are sampled from a mixture of two elliptical distributions with *known* covariance matrix $\Sigma$ and *known* class conditional means $\mu_1$ and $\mu_2$. Shao et al. (2011) and Cai & Liu (2011) leveraged this fact to show that, under certain conditions on the sparsity of either $\mu_1 - \mu_2$ and $\Sigma$ or the sparsity of $\Sigma^{-1}(\mu_1 - \mu_2)$, both SLDA and AdaLDA classifiers also achieve the Bayes error rate for elliptical distributions with unknown $\Sigma, \mu_1$ and $\mu_2$. We now discuss how the theoretical properties for lda ∘ pca can be extended to elliptical distributions, provided that they have sub-Gaussian tails as define below.

*Definition* 1. Let $\Psi : [0, \infty) \to [0, \infty)$ be a non-decreasing and non-zero convex function with $\Psi(0) = 0$. Let $Z$ be a mean 0 random variable. The Birnbaum-Orlicz $\Psi$-norm of $Z$ is defined as

$$\|Z\|_\Psi = \inf\left\{s \geq 0 : \mathbb{E}\Psi\left(\frac{Z}{s}\right) \leq 2\right\}. \tag{7.3}$$

Similarly, if $\boldsymbol{Z}$ is a mean 0 random vector taking values in $\mathbb{R}^p$ then its $\Psi$-norm is defined as

$$\|\boldsymbol{Z}\|_\Psi = \sup_{w \in \mathbb{R}^p, \|w\|=1} \|w^\top \boldsymbol{Z}\|_\Psi. \tag{7.4}$$

Let $\Psi_1(x) = \exp(|x|)$ and $\Psi_2(x) = \exp(x^2)$. A mean 0 random vector $Z$ is said to be sub-exponential if $\|\boldsymbol{Z}\|_{\Psi_1} < \infty$ and is said to be sub-Gaussian if $\|\boldsymbol{Z}\|_{\Psi_2} < \infty$. If $\boldsymbol{Z} \in \mathbb{R}^p$ is a mean 0 sub-Gaussian random vector then $w^\top \boldsymbol{Z}$ is a sub-Gaussian random variable for all $w \in \mathbb{R}^p$. Furthermore, if $Z$ is a sub-Gaussian random variable then there exists a universal constant $K$ such that for all $t > 0$, we have

$$\mathbb{P}(Z > t) \leq 2\exp\left(-t^2/(K\|Z\|_{\Psi_2})^2\right).$$

For more on sub-Gaussian random vectors, see Vershynin (2018, Section 2.5, Section 3.4).

*Remark* 7. If $\boldsymbol{Z} \in \mathbb{R}^p$ is a mean 0 sub-Gaussian random vector with covariance matrix $\Sigma$ then $\|w^\top \boldsymbol{Z}\|_{\Psi_2}^2 \geq w^\top \Sigma w$ for all $w \in \mathbb{R}^p$. In this paper we shall assume that a converse inequality also holds, namely that there exist a constant $c_1 > 0$ such that,

$$w^\top \Sigma w \geq c_1 \|w^\top \boldsymbol{Z}\|_{\Psi_2}^2, \qquad \text{for all } w \in \mathbb{R}^p. \tag{7.5}$$

We note that the constant $c_1$ in Eq. (7.5) can depend on $\boldsymbol{Z}$ but does not depend on the choice of $w \in \mathbb{R}^p$. If $\boldsymbol{Z}$ is multivariate normal then Eq. (7.5) always hold. If $\boldsymbol{Z}$ is a zero mean sub-Gaussian random vector but not a multivariate normal then Eq. (7.5) allows us to bound the Orlicz norm of $w^\top \boldsymbol{Z}$ for any $w \in \mathbb{R}^p$ in terms of its variance. This then allow us to obtain better estimate for $\hat{\Sigma} - \Sigma$ in spectral norm, especially in the setting where the spiked eigenvalues could diverge with $p$. See for example Theorem 4.7.1 in Vershynin (2018) and Theorem 9 in Koltchinskii & Lounici (2017).

We can now reformulate the classification problem in the earlier part of this paper to the case of elliptical distributions as follows. Let $\{\epsilon_{11}, \ldots, \epsilon_{1n_1}\}$ and $\{\epsilon_{21}, \ldots, \epsilon_{2n_2}\}$ be independently and identically distributed mean 0 random vectors with probability density functions of the form in Eq. (7.2) and suppose that the training sample is given by $X_{ij} = \mu_i + \epsilon_{ij}$ for $i \in \{1, 2\}$ and $j \in \{1, 2, \ldots, n_i\}$.

Given these training samples $\{X_{ij}\}$, let $\mathbf{X}$ be the $(n_1 + n_2) \times p$ matrix whose rows are the $\{X_{ij}\}$. Then Theorem 1, in particular Eq. (3.4), also holds for these $\mathbf{X}$ as long as the $\{\epsilon_{ij}\}$ satisfy Eq. (7.5). This then implies that Theorem 2, in particular Eq. (3.5), also holds when the $\mathbf{X}$ are sub-Gaussian random vectors. The resulting bound for $\|\hat{\zeta} - \zeta\|_\infty$ allows us to recover $\mathcal{S}_\zeta$ by thresholding $\hat{\zeta}$, and hence $\hat{R}_{\text{lda}\circ\text{pca}} \to R_F$. In summary, lda ∘ pca is asymptotically Bayes optimal whenever the feature vectors $\{X_{ij}\}$ are sampled from a mixture of elliptical distributions with sub-Gaussian tails.

# 8 Proofs of Stated Results

This section contains the proofs of Theorem 1 through Theorem 3. We will present the proofs under the more general assumption that the feature vectors $\{X_{ij}\}$ are sub-Gaussian random vectors; see also the discussion in Section 7.3.

## 8.1 Preliminary results

We start by listing some elementary but useful facts about the $2 \to \infty$ norm and its relationships with other matrix norms. Recall that $\|\cdot\|$ denote the spectral norm if its argument is a matrix and denote the $\ell_2$ norm if its argument is a vector.

**Proposition 1.** *Let $A \in \mathbb{R}^{p_1 \times p_2}$ and $B \in \mathbb{R}^{p_2 \times p_3}$ be arbitrary real-valued matrices. Let $x \in \mathbb{R}^{p_2}$ be an arbitrary vector. For a given $i \in [p_1]$, let $\boldsymbol{e}_i$ denote the ith elementary basis vector in $\mathbb{R}^{p_1}$. Then*

$$\|A\|_{2\to\infty} = \max_{i\in[p_1]} \|A^\top \mathbf{e}_i\|; \tag{8.1}$$

$$\|Ax\|_\infty \le \|A\|_{2\to\infty} \times \|x\|; \tag{8.2}$$

$$\|AB\|_{2\to\infty} \le \|A\|_{2\to\infty} \times \|B\|. \tag{8.3}$$

Eq. (8.1) states that the two-to-infinity norm of a matrix $A$ is equivalent to the *maximum* $\ell_2$ norm of the rows of $A$. Eq. (8.2) provides a bound for $\|Ax\|_\infty$ that is tighter than the naive bound $\|Ax\|_\infty \le \|Ax\| \le \|A\|\,\|x\|$.

Throughout the section, we will make use of Bernstein's inequality. For completeness, we state the result below without proof.

**Proposition 2.** *Let $X_1, X_2, \ldots, X_n$ be independent, mean-zero, sub-exponential random variables. Then for all $t > 0$, there exists a constant $c > 0$ such that:*

$$\mathbb{P}(|\sum_{i=1}^{n} X_i| \ge t) \le 2\exp\left(-c \cdot \min\left(\frac{t^2}{\sum_{i=1}^{n}\|X_i\|_{\psi_1}^2}, \frac{t}{\max_i \|X_i\|_{\psi_1}}\right)\right). \tag{8.4}$$

*where $\|\cdot\|_{\psi_1}$ denotes the sub-exponential Orlicz norm.*

Let $\xi_i = \bar{X}_i - \mu_i$. Recall that $\eta_k = (\lambda_k + \sigma^2)^{-1/2}$ and $\sigma^2 = \frac{1}{p-d}(\mathrm{tr}(\Sigma) - \mathrm{tr}(\Lambda))$ are the eigenvalues of $\mathcal{W} = \Sigma^{-1/2}$. The following lemma provides several concentration inequalities for $\hat{\Sigma} - \Sigma$, $\hat{\eta}_k - \eta_k$ and $\sigma^2 - \hat{\sigma}^2$.

**Lemma 1.** *Assume that the random variables $\{X_i\}$ satisfy Eq. (7.5) and the covariance matrix $\Sigma$ satisfies Assumption 3 and Assumption 4. Then the following bounds hold simultaneously with probability at least $1 - p^{-2}$ (where $\hat{\Sigma}_0$ is defined in Eq. (2.4))*

$$\|\hat{\Sigma}_0 - \Sigma\| = \mathcal{O}(p\sqrt{n^{-1}\ln p}), \tag{8.5}$$

$$\|\xi_i\|^2 = \mathcal{O}(p\,n^{-1}\ln p), \tag{8.6}$$

$$\|\hat{\Sigma} - \Sigma\| = \mathcal{O}(p\sqrt{n^{-1}\ln p}), \tag{8.7}$$

$$|\hat{\eta}_k - \eta_k| = \mathcal{O}(\sqrt{p^{-1}n^{-1}\ln p}), \quad \text{for all } k \in [d] \tag{8.8}$$

$$|\sigma^2 - \hat{\sigma}^2| = \mathcal{O}(\sqrt{n^{-1}\ln p}). \tag{8.9}$$

Eq. (8.5) is given in Lounici (2014) and Koltchinskii & Lounici (2017) while Eq. (8.6) follows from an application of Bernstein inequality; see Section 2 and Section 3 of Vershynin (2018). Eq. (8.7) follows from Eq. (8.5) and Eq. (8.6) together with the observation that $\hat{\Sigma}_0 - \hat{\Sigma} = n^{-1}(n_1\xi_1\xi_1^\top + n_2\xi_2\xi_2^\top)$. Finally, Eq. (8.8) and Eq. (8.9) follow from Eq. (8.7) and Weyl's inequality.

## 8.2 Proof of Theorem 1

Recall that $\mathcal{U}$ and $\hat{\mathcal{U}}$ denote the $p \times d$ matrices whose columns are the orthonormal eigenvectors corresponding to the $d$ largest eigenvalues of $\Sigma$ and the *pooled* sample covariance matrix $\hat{\Sigma}$, respectively. Now let $\mathcal{U}_\perp$ and

$\hat{\mathcal{U}}_\perp$ be the $p \times (p-d)$ matrices whose orthonormal columns are the remaining eigenvectors of $\Sigma$ and $\hat{\Sigma}$, respectively, i.e., $\mathcal{I}_p - \mathcal{U}\mathcal{U}^\top = \mathcal{U}_\perp \mathcal{U}_\perp^\top$ and $\mathcal{I}_p - \hat{\mathcal{U}}\hat{\mathcal{U}}^\top = \hat{\mathcal{U}}_\perp \hat{\mathcal{U}}_\perp^\top$.

Let $\Xi$ be the $d \times d$ orthogonal matrix that minimizes

$$\min_W \|W - \mathcal{U}^\top \hat{\mathcal{U}}\|_F$$

among all orthogonal matrices. Let $\mathcal{E}_n = \hat{\Sigma} - \Sigma$. Then by Theorem 3.7 in Cape et al. (2019) we have

$$
\begin{aligned}
\|\hat{\mathcal{U}} - \mathcal{U}\Xi\|_{2\to\infty} \leq\ & 2(\lambda_d + \sigma^2)^{-1} \|(\mathcal{U}_\perp \mathcal{U}_\perp^\top)\mathcal{E}_n(\mathcal{U}\mathcal{U}^\top)\|_{2\to\infty} \\
& + 2(\lambda_d + \sigma^2)^{-1} \|(\mathcal{U}_\perp \mathcal{U}_\perp^\top)\mathcal{E}_n(\mathcal{U}_\perp \mathcal{U}_\perp^\top)\|_{2\to\infty} \times \|\sin\Theta(\hat{\mathcal{U}},\mathcal{U})\| \\
& + 2(\lambda_d + \sigma^2)^{-1} \|(\mathcal{U}_\perp \mathcal{U}_\perp^\top)\Sigma(\mathcal{U}_\perp \mathcal{U}_\perp^\top)\|_{2\to\infty} \times \|\sin\Theta(\hat{\mathcal{U}},\mathcal{U})\| \\
& + \|\sin\Theta(\hat{\mathcal{U}},\mathcal{U})\|^2 \times \|\mathcal{U}\|_{2\to\infty}.
\end{aligned}
\tag{8.10}
$$

Now recall the matrix $\hat{\Sigma}_0$ from Eq. (2.4). We then have

$$
\underbrace{\hat{\Sigma} - \Sigma}_{\mathcal{E}_n} = \underbrace{\hat{\Sigma}_0 - \Sigma}_{E_n} - \frac{n_1}{n}\underbrace{(\bar{x}_1 - \mu_1)(\bar{x}_1 - \mu_1)^\top}_{\mathrm{E}_1} - \frac{n_2}{n}\underbrace{(\bar{x}_2 - \mu_2)(\bar{x}_2 - \mu_2)^\top}_{\mathrm{E}_2}.
$$

Using the same argument as that for the proof of Theorem 1.1 in Cape et al. (2019) we have with probability at least $1 - p^{-2}$ that

$$\|\mathcal{U}_\perp \mathcal{U}_\perp E_n \mathcal{U}\mathcal{U}^\top\|_{2\to\infty} \leq \mathcal{C}d\left(\max_{i\in[p]} \Sigma_{ii}\right)^{1/2} \times \sqrt{\frac{(\lambda_1 + \sigma^2)\ln p}{n}} \tag{8.11}$$

$$\|\mathcal{U}_\perp \mathcal{U}_\perp^\top E_n \mathcal{U}_\perp \mathcal{U}_\perp^\top\|_{2\to\infty} \leq \mathcal{C}\sigma\sqrt{\frac{(\lambda_1 + \sigma^2)\ln p}{n}} \tag{8.12}$$

where we have used the assumption that $\lambda_k = \Theta(p)$ for all $k \in [d]$ so that $\boldsymbol{r}(\Sigma)$ – the effective rank of $\Sigma$ – is bounded. Here and in the subsequent derivations we will, for simplicity of presentation, use $\mathcal{C}$ to denote a finite and *universal* constant that can change from line to line.

Therefore to complete the proof of Theorem 1, it suffices to show that, for $j \in \{1,2\}$, the terms $\|\mathcal{U}_\perp \mathcal{U}_\perp \mathrm{E}_j \mathcal{U}\mathcal{U}^\top\|_{2\to\infty}$ and $\|\mathcal{U}_\perp \mathcal{U}_\perp^\top \mathrm{E}_j \mathcal{U}_\perp \mathcal{U}_\perp^\top\|_{2\to\infty}$ are of the same or smaller order than those in Eqs. (8.11) and (8.12), respectively.

We now bound $\|\mathcal{U}_\perp \mathcal{U}_\perp \mathrm{E}_1 \mathcal{U}\mathcal{U}^\top\|_{2\to\infty}$. From Assumption 4 and Proposition 1 we have

$$\|\mathcal{U}_\perp \mathcal{U}_\perp^\top\|_\infty \leq \mathcal{C}\sqrt{d}, \quad \text{and} \quad \|\mathcal{U}_\perp \mathcal{U}_\perp^\top \mathrm{E}_1 \mathcal{U}\mathcal{U}^\top\|_{2\to\infty} \leq \mathcal{C}\sqrt{d}\|\mathrm{E}_1\mathcal{U}\|_{2\to\infty}. \tag{8.13}$$

Furthermore, we also have

$$\|\mathrm{E}_1 \mathcal{U}\|_{2\to\infty} \leq \sqrt{d} \max_{i\in[p],j\in[d]} \left|\langle \mathrm{E}_1 \mathbf{e}_i^{(p)}, \mathbf{u}_j\rangle\right| = \sqrt{d}\max_{i\in[p],j\in[d]}\left|[(\bar{X}_1 - \mu_1)^\top \mathbf{e}_i^{(p)}] \times [(\bar{X}_1 - \mu_1)^\top \mathbf{u}_j]\right|.$$

Since $X_i$ is sub-Gaussian, by the properties of Orlicz norms we have

$$\left\|[(\bar{X}_1 - \mu_1)^\top \mathbf{e}_i^{(p)}] \times [(\bar{X}_1 - \mu_1)^\top \mathbf{u}_j]\right\|_{\Psi_1} \leq \left\|(\bar{X}_1 - \mu_1)^\top \mathbf{e}_i^{(p)}\right\|_{\Psi_2} \times \left\|(\bar{X}_1 - \mu_1)^\top \mathbf{u}_j\right\|_{\Psi_2}$$

Eq. (7.5) implies that there exists a constant $\mathcal{C} > 0$ such that for any $i \in [p]$ and $j \in [d]$,

$$\left\|(\bar{X}_1 - \mu_1)^\top \mathbf{e}_i^{(p)}\right\|_{\Psi_2} \leq \mathcal{C}\sqrt{\frac{\Sigma_{ii}}{n_1}} \leq \frac{\mathcal{C}}{\sqrt{n_1}}\left(\max_{i\in[p]}\Sigma_{ii}\right)^{1/2}, \tag{8.14}$$

$$\left\|(\bar{X}_1 - \mu_1)^\top \mathbf{u}_j\right\|_{\Psi_2} \leq \mathcal{C}\sqrt{\mathrm{Var}(\boldsymbol{u}_j^\top(\bar{X}_1 - \mu_1))} \leq \mathcal{C}\sqrt{\boldsymbol{u}_j^\top n^{-1}\Sigma \boldsymbol{u}_j} \leq \mathcal{C}\sqrt{\frac{\lambda_1 + \sigma^2}{n_1}}. \tag{8.15}$$

Now fix an arbitrary pair $(i, j)$ with $i \in [p]$ and $j \in [d]$. Then by Assumption 4, Eqs. (8.14) and (8.15), and properties of sub-exponential random variables, we have

$$\mathbb{E}[|\langle \mathrm{E}_1 \mathbf{e}_i^{(p)}, \mathbf{u}_j \rangle|] \leq \frac{\mathcal{C}(\lambda_1 + \sigma^2)}{n_1} \times \sqrt{\frac{d}{p}}.$$

Furthermore, by Bernstein inequality (Vershynin, 2018, Section 2.8), there exists a constant $\mathcal{C} > 0$ such that with probability at least $1 - \mathcal{O}(p^{-3})$,

$$|\langle \mathrm{E}_1 \mathbf{e}_i^{(p)}, \mathbf{u}_j \rangle| \leq \ \mathbb{E}[|\langle \mathrm{E}_1 \mathbf{e}_i^{(p)}, \mathbf{u}_j \rangle|] + \frac{\mathcal{C} \ln p}{n_1} \Big(\max_{i \in [p]} \Sigma_{ii}\Big)^{1/2} \sqrt{\lambda_1 + \sigma^2}. \tag{8.16}$$

Now recall Eq. (8.13). Then by Eq. (8.16) together with a union bound over all $i \in [p]$ and $j \in [d]$ we have, with probability at least $1 - \mathcal{O}(p^{-2})$,

$$\|\mathcal{U}_\perp \mathcal{U}_\perp \mathrm{E}_1 \mathcal{U} \mathcal{U}\|_{2 \to \infty} \leq \mathcal{C} \sqrt{d} \|\mathrm{E}_1 \mathcal{U}\|_{2 \to \infty} \leq \mathcal{C} d \max_{i,j} |\langle \mathrm{E}_1 \mathbf{e}_i^{(p)}, \mathbf{u}_j \rangle| \leq \mathcal{C} \frac{d^{3/2} \sqrt{p} \ln p}{n}. \tag{8.17}$$

where we had used Assumption 3, namely $n_1 \asymp n$ and $\lambda_1 \asymp p$, when simplifying the above expression. An almost identical argument also yields

$$\|\mathcal{U}_\perp \mathcal{U}_\perp \mathrm{E}_2 \mathcal{U} \mathcal{U}\|_{2 \to \infty} \leq \mathcal{C} \frac{d^{3/2} \sqrt{p} \ln p}{n}$$

with probability at least $1 - \mathcal{O}(p^{-2})$. We therefore have

$$\begin{aligned}
\frac{2\|\mathcal{U}_\perp \mathcal{U}_\perp^\top \mathcal{E}_n \mathcal{U} \mathcal{U}^\top\|_{2 \to \infty}}{\lambda_d + \sigma^2} &\leq \frac{\mathcal{C}}{\lambda_d + \sigma^2} \Big( d\Big(\max_i \Sigma_{ii}\Big)^{1/2} \times \sqrt{\frac{(\lambda_1 + \sigma^2) \ln p}{n}} + \frac{d^{3/2} \sqrt{p} \ln p}{n} \Big) \\
&\leq \mathcal{C} d^{3/2} \sqrt{\frac{\ln p}{np}}
\end{aligned} \tag{8.18}$$

We next consider $\|\mathcal{U}_\perp \mathcal{U}_\perp^\top \mathrm{E}_1 \mathcal{U}_\perp \mathcal{U}_\perp^\top\|_{2 \to \infty}$. For $j > d$, we have

$$\|(\bar{X}_1 - \mu_1)^\top \boldsymbol{u}_j\|_{\Psi_2} \leq \mathcal{C} \sqrt{\boldsymbol{u}_j^\top n^{-1} \Sigma \boldsymbol{u}_j} \leq \mathcal{C} n^{-1/2} \sigma.$$

Then following the same argument as that used for showing Eq. (8.17), we have

$$\big\|\mathcal{U}_\perp \mathcal{U}_\perp^\top \mathrm{E}_1 \mathcal{U}_\perp \mathcal{U}_\perp^\top\big\|_{2 \to \infty} \leq \mathcal{C} \sqrt{pd} \, \sigma \Big(\max_{i \in [p]} \Sigma_{ii}\Big)^{1/2} \times \frac{\ln p}{n_1} \leq \mathcal{C} \sigma \sqrt{pd} \, \frac{\ln p}{n} \tag{8.19}$$

with probability at least $1 - \mathcal{O}(p^{-2})$, and similarly for $\|\mathcal{U}_\perp \mathcal{U}_\perp^\top \mathrm{E}_2 \mathcal{U}_\perp \mathcal{U}_\perp^\top\|_{2 \to \infty}$.

Next we have, by the Davis-Kahan theorem and Lemma 1, that

$$\|\sin \Theta(\hat{\mathcal{U}}, \mathcal{U})\| \leq \frac{\|\mathcal{E}_n\|}{\lambda_d + \sigma^2} \leq \mathcal{C} \sqrt{\frac{\ln p}{n}} \tag{8.20}$$

with probability at least $1 - \mathcal{O}(p^{-2})$. From Eqs. (8.12), (8.19) and (8.20), together with a similar argument as that for showing Eq. (8.18), we have

$$\frac{2\|\mathcal{U}_\perp \mathcal{U}_\perp^\top \mathcal{E}_n \mathcal{U}_\perp \mathcal{U}_\perp^\top\|_{2 \to \infty} \times \|\sin \Theta(\hat{\mathcal{U}}, \mathcal{U})\|}{(\lambda_d + \sigma^2)} \leq \mathcal{C} \sigma d \times \frac{\ln p}{n \sqrt{p}} \tag{8.21}$$

with probability at least $1 - \mathcal{O}(p^{-2})$. Eq. (8.20) together with Assumption 4 also imply

$$\|\sin \Theta(\hat{\mathcal{U}}, \mathcal{U})\|_2^2 \times \|\mathcal{U}\|_{2 \to \infty} \leq \frac{\mathcal{C} \ln p}{n} \times \frac{\sqrt{d}}{\sqrt{p}} = \frac{\mathcal{C} \sqrt{d} \ln p}{n \sqrt{p}}. \tag{8.22}$$

with probability at least $1 - \mathcal{O}(p^{-2})$. Next note that

$$\|\mathcal{U}_\perp \mathcal{U}_\perp^\top \Sigma \mathcal{U}_\perp \mathcal{U}_\perp^\top\|_{2\to\infty} = \|\sigma^2 \mathcal{U}_\perp \mathcal{U}_\perp^\top\|_{2\to\infty} \le \sigma^2 \|\mathcal{U}_\perp \mathcal{U}_\perp^\top\| = \sigma^2.$$

We therefore have

$$\frac{2\|\mathcal{U}_\perp \mathcal{U}_\perp^\top \Sigma \mathcal{U}_\perp \mathcal{U}_\perp^\top\|_{2\to\infty} \times \|\sin\Theta(\hat{\mathcal{U}}, \mathcal{U})\|}{\lambda_d + \sigma^2} \le \frac{\mathcal{C}\sigma^2}{\lambda_d + \sigma^2} \times \sqrt{\frac{\ln p}{n}} \le \frac{\mathcal{C}\sigma^2 \sqrt{\ln p}}{\sqrt{n}p} \tag{8.23}$$

Substituting the bounds in Eqs. (8.18), (8.21), (8.22) and (8.23) into Eq. (8.10) we obtain

$$\|\hat{\mathcal{U}} - \mathcal{U}\Xi\|_{2\to\infty} \le \mathcal{C}\Big(\frac{d^{3/2}\sqrt{\ln p}}{\sqrt{n}p} + \frac{d\ln p}{n\sqrt{p}} + \frac{\sigma^2 \sqrt{\ln p}}{\sqrt{n}p}\Big) \le \mathcal{C}\sqrt{\frac{d^3 \ln p}{np}}$$

with probability at least $1 - p^{-2}$. This completes the proof of Theorem 1.

### 8.3 Proof of Theorem 2

First recall that $\hat{\zeta} = \hat{\mathcal{W}}(\bar{X}_2 - \bar{X}_1)$ and $\zeta = \mathcal{W}(\mu_2 - \mu_1)$ where the whitening matrix $\mathcal{W}$ and its estimate $\hat{\mathcal{W}}$ are given by Eq. (3.2) and Eq. (3.3), respectively. We now consider the decomposition

$$\hat{\zeta} - \zeta = \underbrace{\mathcal{W}\big[(\bar{X}_2 - \bar{X}_1) - (\mu_2 - \mu_1)\big]}_{A} + \underbrace{(\hat{\mathcal{W}} - \mathcal{W})\big[(\bar{X}_2 - \bar{X}_1) - (\mu_2 - \mu_1)\big]}_{B} + \underbrace{(\hat{\mathcal{W}} - \mathcal{W})(\mu_2 - \mu_1)}_{C}.$$

We will now bound each of the term in the right hand side of the above display. We start with the term in $(A)$. Let $\delta = (\bar{X}_2 - \bar{X}_1) - (\mu_2 - \mu_1)$ and let $\xi = \mathcal{W}\delta$. We then have $\mathbb{E}[\xi] = \mathbf{0}$ and $\mathrm{Var}[\xi] = c\,n^{-1}\mathcal{I}_p$ for some finite constant $c$. Since $\delta$ satisfies Eq. (7.5), $\xi$ also satisfies Eq. (7.5). Hence, by Bernstein inequality for sub-Gaussian random vectors, there exists a constant $\mathcal{C} > 0$ such that with probability at least $1 - \mathcal{O}(p^{-2})$,

$$\big\|\mathcal{W}\big((\bar{X}_2 - \bar{X}_1) - (\mu_2 - \mu_1)\big)\big\|_\infty = \|\xi\|_\infty \le \mathcal{C}\sqrt{\frac{\ln p}{n}}. \tag{8.24}$$

We now bound the terms in $(B)$ and $(C)$. Let $\hat{\mathcal{D}}$ and $\mathcal{D}$ be diagonal matrices where

$$\hat{\mathcal{D}} = \big(\hat{\Lambda} + \hat{\sigma}^2 \mathcal{I}_d\big)^{-1/2}, \quad \mathcal{D} = \big(\Lambda + \sigma^2 \mathcal{I}_d\big)^{-1/2}.$$

We start by decomposing $\hat{\mathcal{W}} - \mathcal{W}$ as

$$\hat{\mathcal{W}} - \mathcal{W} = \underbrace{\hat{\mathcal{U}}\hat{\mathcal{D}}\hat{\mathcal{U}}^\top - \mathcal{U}\mathcal{D}\mathcal{U}^\top}_{I} + \underbrace{(\hat{\sigma}^{-1} - \sigma^{-1})(\mathcal{I}_p - \mathcal{U}\mathcal{U}^\top)}_{II} + \underbrace{\hat{\sigma}^{-1}(\hat{\mathcal{U}}\hat{\mathcal{U}}^\top - \mathcal{U}\mathcal{U}^\top)}_{III}. \tag{8.25}$$

Now consider the term $(\hat{\sigma}^{-1} - \sigma^{-1})(\mathcal{I}_p - \mathcal{U}\mathcal{U}^\top)\delta$ obtained by combining the expressions in $(B)$ and $(II)$. The covariance matrix for $\delta$ is $n^{-1}\Sigma$ and hence $(\mathcal{I}_p - \mathcal{U}\mathcal{U})^\top \delta$ satisfies Eq. (7.5) with covariance matrix $n^{-1}\sigma^2(\mathcal{I}_p - \mathcal{U}\mathcal{U}^\top)$. Therefore, by Bernstein inequality, there exists a constant $\mathcal{C} > 0$ such that with probability at least $1 - \mathcal{O}(p^{-2})$,

$$\|(\mathcal{I}_p - \mathcal{U}\mathcal{U}^\top)\delta\|_\infty \le \mathcal{C}\sigma\sqrt{\frac{\ln p}{n}}.$$

Furthermore, from Lemma 1, we have with probability at least $1 - \mathcal{O}(p^{-2})$ that

$$|\hat{\sigma}^{-1} - \sigma^{-1}| = \frac{|\hat{\sigma}^2 - \sigma^2|}{\hat{\sigma}\sigma(\hat{\sigma} + \sigma)} \le \mathcal{C}\sqrt{\frac{\ln p}{n}}. \tag{8.26}$$

Combining the above bounds, we obtain

$$\|(\hat{\sigma}^{-1} - \sigma^{-1})(\mathcal{I} - \mathcal{U}\mathcal{U}^\top)\delta\|_\infty \le \frac{C\ln p}{n}. \tag{8.27}$$

with probability at least $1 - \mathcal{O}(p^{-2})$.

We next consider the term $(\hat{\sigma}^{-1} - \sigma^{-1})(\mathcal{I}_p - \mathcal{U}\mathcal{U}^\top)(\mu_2 - \mu_1)$. Recall that if $\mathcal{U}$ has bounded coherence as in Assumption 4 then $\|\mathcal{I}_p - \mathcal{U}\mathcal{U}^\top\|_\infty \leq (1 + \mathcal{C}_\mathcal{U})\sqrt{d}$ where $\mathcal{C}_\mathcal{U}$ is a finite constant. We therefore have, by Lemma 1, that

$$
\begin{aligned}
\|(\hat{\sigma}^{-1} - \sigma^{-1})(\mathcal{I}_p - \mathcal{U}\mathcal{U}^\top)(\mu_2 - \mu_1)\|_\infty &= \|(\hat{\sigma}^{-1} - \sigma^{-1})(\mathcal{I}_p - \mathcal{U}\mathcal{U}^\top)\Sigma^{1/2}\mathcal{W}(\mu_2 - \mu_1)\|_\infty \\
&= \|(\hat{\sigma}^{-1} - \sigma^{-1})\sigma(\mathcal{I}_p - \mathcal{U}\mathcal{U})^\top \zeta\|_\infty \\
&\leq |(\hat{\sigma}^{-1} - \sigma^{-1})| \times \sigma(1 + \mathcal{C}_\mathcal{U})\sqrt{d} \times \|\zeta\|_\infty \\
&\leq \mathcal{C}\sqrt{\frac{\ln p}{n}} \times \|\zeta\|_\infty,
\end{aligned}
$$

with probability at least $1 - \mathcal{O}(p^{-2})$.

We now focus our efforts on terms involving $\hat{\mathcal{U}}\hat{\mathcal{D}}\hat{\mathcal{U}}^\top - \mathcal{U}\mathcal{D}\mathcal{U}^\top$. Let $\Xi$ be the minimizer of $\|W - \mathcal{U}^\top \hat{\mathcal{U}}\|_F$ among all $d \times d$ orthogonal matrices $W$. We then have

$$
\begin{aligned}
\hat{\mathcal{U}}\hat{\mathcal{D}}\hat{\mathcal{U}}^\top - \mathcal{U}\mathcal{D}\mathcal{U}^\top &= (\hat{\mathcal{U}} - \mathcal{U}\mathcal{U}^\top\hat{\mathcal{U}})\hat{\mathcal{D}}\hat{\mathcal{U}}^\top + \mathcal{U}\mathcal{U}^\top\hat{\mathcal{U}}\hat{\mathcal{D}}\hat{\mathcal{U}}^\top - \mathcal{U}\mathcal{D}\mathcal{U}^\top \\
&= \underbrace{\left[(\hat{\mathcal{U}} - \mathcal{U}\Xi) - \mathcal{U}(\mathcal{U}^\top\hat{\mathcal{U}} - \Xi)\right]\hat{\mathcal{D}}\hat{\mathcal{U}}^\top}_{\text{Part 1}} + \underbrace{\mathcal{U}\mathcal{U}^\top\hat{\mathcal{U}}\hat{\mathcal{D}}\hat{\mathcal{U}}^\top - \mathcal{U}\mathcal{D}\mathcal{U}^\top}_{\text{Part 2}}
\end{aligned}
$$

We now note a few elementary but useful algebraic facts frequently used in the subsequent derivations.

*Fact* 1.

$$\|\mathcal{U}^\top\hat{\mathcal{U}} - \Xi\| \leq \|\sin\Theta(\hat{\mathcal{U}}, \mathcal{U})\|^2, \tag{8.28}$$

$$\|\mathcal{U}\mathcal{U}^\top\hat{\mathcal{U}}_\perp\hat{\mathcal{U}}_\perp^\top\| = \|\hat{\mathcal{U}}\hat{\mathcal{U}}^\top\mathcal{U}_\perp\mathcal{U}_\perp^\top\| = \|\mathcal{U}^\top\hat{\mathcal{U}}_\perp\hat{\mathcal{U}}_\perp^\top\| = \|\hat{\mathcal{U}}^\top\mathcal{U}_\perp\mathcal{U}_\perp^\top\| = \|\sin\Theta(\hat{\mathcal{U}}, \mathcal{U})\|, \tag{8.29}$$

$$\|\sin\Theta(\mathcal{U}, \hat{\mathcal{U}})\| \leq \|\hat{\mathcal{U}}\hat{\mathcal{U}}^\top - \mathcal{U}\mathcal{U}^\top\| \leq 2\|\sin\Theta(\hat{\mathcal{U}}, \mathcal{U})\|, \tag{8.30}$$

$$\|\mathcal{U}^\top\hat{\mathcal{U}}\hat{\mathcal{D}} - \mathcal{D}\mathcal{U}^\top\hat{\mathcal{U}}\| \leq \|\mathcal{U}^\top(\Sigma - \hat{\Sigma})\hat{\mathcal{U}}\| \times \|\mathbb{H}\|. \tag{8.31}$$

where $\mathbb{H} = (\mathbb{H}_{ij})$ is a $d \times d$ matrix with entries

$$\mathbb{H}_{ij} = \frac{1}{\sqrt{(\hat{\lambda}_j + \hat{\sigma}^2)(\lambda_i + \sigma^2)}(\sqrt{\lambda_i + \hat{\sigma}^2} + \sqrt{\hat{\lambda}_j + \sigma^2})}.$$

Eq. (8.28) is from Lemma 6.7 in Cape et al. (2019) while Eqs. (8.29) and (8.30) are standard results for the sin-$\Theta$ distance (see for example Lemma 1 in Cai & Zhang (2018)). Finally, Eq. (8.31) follows from the observation

$$
\begin{aligned}
(\mathcal{U}^\top\hat{\mathcal{U}}\hat{\mathcal{D}} - \mathcal{D}\mathcal{U}^\top\hat{\mathcal{U}})_{ij} &= (\mathcal{U}^\top\hat{\mathcal{U}})_{i,j}(\hat{\mathcal{D}}_{jj} - \mathcal{D}_{ii}) \\
&= (\mathcal{U}^\top\hat{\mathcal{U}})_{ij} \frac{(\lambda_i + \sigma^2) - (\hat{\lambda}_j + \hat{\sigma}^2)}{(\hat{\lambda}_j + \hat{\sigma}^2)^{1/2}(\lambda_i + \sigma^2)^{1/2}\big((\hat{\lambda}_j + \hat{\sigma}^2)^{1/2} + (\lambda_i + \sigma^2)^{1/2}\big)} \\
&= (\Lambda\mathcal{U}^\top\hat{\mathcal{U}} - \mathcal{U}^\top\hat{\mathcal{U}}\hat{\Lambda})_{ij}\mathbb{H}_{ij} = (\mathcal{U}^\top(\Sigma - \hat{\Sigma})\hat{\mathcal{U}})_{ij}\mathbb{H}_{ij}
\end{aligned}
$$

We thus have $\mathcal{U}^\top\hat{\mathcal{U}}\hat{\mathcal{D}} - \mathcal{D}\mathcal{U}^\top\hat{\mathcal{U}} = (\mathcal{U}^\top(\Sigma - \hat{\Sigma})\hat{\mathcal{U}}) \circ \mathbb{H}$. Therefore, by Schur inequality for Hadamard product (see e.g., Theorem 5.5.1 of Horn & Johnson (1991)), we have

$$\|(\mathcal{U}^\top(\Sigma - \hat{\Sigma})\hat{\mathcal{U}}) \circ \mathbb{H}\| \leq \|(\mathcal{U}^\top(\Sigma - \hat{\Sigma})\hat{\mathcal{U}})\| \times \|\mathbb{H}\|.$$

We next state a technical lemma for bounding several terms that appears frequently in our analysis.

**Lemma 2.** *Suppose that Assumption 1 through Assumption 4 are satisfied. Then with probability at least* $1 - \mathcal{O}(p^{-2})$, *the following bounds hold simultaneously*

$$\|\mathbb{H}\| = \mathcal{O}(d^{3/2}p^{-3/2}), \tag{8.32}$$

$$\|\mathcal{U}^{\top}(\Sigma - \hat{\Sigma})\mathcal{U}\| = \mathcal{O}(p\sqrt{n^{-1}\ln p}), \tag{8.33}$$

$$\|\mathcal{U}^{\top}(\Sigma - \hat{\Sigma})\hat{\mathcal{U}}\| = \mathcal{O}(p\sqrt{n^{-1}\ln p}), \tag{8.34}$$

$$\|\mathcal{U}^{\top}\hat{\mathcal{U}}\hat{\mathcal{D}} - \mathcal{D}\mathcal{U}^{\top}\hat{\mathcal{U}}\| = \mathcal{O}(n^{-1/2}p^{-1/2}\ln p), \tag{8.35}$$

$$\|\hat{\mathcal{U}}\hat{\mathcal{U}}^{\top} - \mathcal{U}\mathcal{U}^{\top}\| = \mathcal{O}(\sqrt{n^{-1}\ln p}). \tag{8.36}$$

Eqs. (8.33) and (8.34) follows from the sub-multiplicativity of the spectral norm together with bounds for $\|\hat{\Sigma}_0 - \Sigma\|$ from Lounici (2014) and Koltchinskii & Lounici (2017). Eq. (8.36) is a consequence of Eq. (8.30) and the Davis-Kahan theorem. Eq. (8.32) follows from Weyl's inequality and the bound for $\|\hat{\Sigma} - \Sigma\|$. Finally, Eqs. (8.31), (8.32) and (8.34) together imply Eq. (8.35).

With the above preparations in place, we now resume our proof of Theorem 2. We first have

$$
\begin{aligned}
\|(\hat{\mathcal{U}} - \mathcal{U}\mathcal{U}^{\top}\hat{\mathcal{U}})\hat{\mathcal{D}}\hat{\mathcal{U}}^{\top}\delta\|_{\infty} &\leq (\|\hat{\mathcal{U}} - \mathcal{U}\Xi\|_{2\to\infty} + \|\mathcal{U}\|_{2\to\infty}\|\mathcal{U}^{\top}\hat{\mathcal{U}} - \Xi\|)\|\hat{\mathcal{D}}\hat{\mathcal{U}}^{\top}\delta\| \\
&\leq \mathcal{C}\Big(\sqrt{\frac{d^3\ln p}{np}} + \frac{\sqrt{d}\ln p}{n\sqrt{p}}\Big)\|\hat{\mathcal{D}}\hat{\mathcal{U}}^{\top}\delta\| \\
&\leq \mathcal{C}\Big(\sqrt{\frac{d^3\ln p}{np}} + \frac{\sqrt{d}\ln p}{n\sqrt{p}}\Big)(\|\hat{\mathcal{D}}\| + \|\hat{\mathcal{D}} - \mathcal{D}\|)\|\hat{\mathcal{U}}^{\top}\delta\| \\
&\leq \mathcal{C}\Big(\sqrt{\frac{d^3\ln p}{np}} + \frac{\sqrt{d}\ln p}{n\sqrt{p}}\Big)\Big(\frac{1}{\sqrt{\lambda_1 + \sigma^2}} + \sqrt{\frac{\ln p}{np}}\Big)\|\hat{\mathcal{U}}^{\top}\delta\| \\
&\leq \frac{\mathcal{C}\sqrt{\ln p}}{p\sqrt{n}}\|\hat{\mathcal{U}}^{\top}\delta\|
\end{aligned}
\tag{8.37}
$$

with probability at least $1 - \mathcal{O}(p^{-2})$. For the above inequality, we have used Lemma 1 to bound $\|\hat{\mathcal{D}} - \mathcal{D}\|$ and used Theorem 1 to bound $\|\hat{\mathcal{U}} - \mathcal{U}\Xi\|_{2\to\infty}$. Finally we used Eq. (8.28), Eq. (8.30) and Eq. (8.36) to bound $\|\hat{\mathcal{U}}^{\top}\mathcal{U} - \Xi\|$.

Next let $T = \mathcal{I}_p - \hat{\mathcal{U}}\hat{\mathcal{U}}^{\top}$. Then

$$
\begin{aligned}
\|(\mathcal{U}\mathcal{U}^{\top}\hat{\mathcal{U}}\hat{\mathcal{D}}\hat{\mathcal{U}} - \mathcal{U}\mathcal{D}\mathcal{U}^{\top})\delta\|_{\infty} &\leq \|\mathcal{U}(\mathcal{U}^{\top}\hat{\mathcal{U}}\hat{\mathcal{D}} - \mathcal{D}\mathcal{U}^{\top}\hat{\mathcal{U}})\hat{\mathcal{U}}^{\top}\delta\|_{\infty} + \|\mathcal{U}\mathcal{D}\mathcal{U}^{\top}T\delta\|_{\infty} \\
&\leq \|\mathcal{U}\|_{2\to\infty}(\|\mathcal{U}^{\top}\hat{\mathcal{U}}\hat{\mathcal{D}} - \mathcal{D}\mathcal{U}^{\top}\hat{\mathcal{U}}\| + \|\mathcal{D}\| \times \|\mathcal{U}^{\top}T\|)\|\delta\| \\
&\leq \Big(\frac{\mathcal{C}\sqrt{d}\ln p}{p\sqrt{n}} + \frac{\mathcal{C}\sqrt{d\ln p}}{p\sqrt{n}}\Big) \times \sqrt{\frac{p\ln p}{n}} \leq \frac{\mathcal{C}\ln p}{n\sqrt{p}}
\end{aligned}
\tag{8.38}
$$

with probability at least $1 - O(p^{-2})$. In the above derivations, we bound $\|\mathcal{U}^{\top}\hat{\mathcal{U}}\hat{\mathcal{D}} - \mathcal{D}\mathcal{U}^{\top}\hat{\mathcal{U}}\|$ using Eq. (8.35), and bound $\|\mathcal{U}^{\top}T\| = \|\mathcal{U}^{\top}(\mathcal{I}_p - \hat{\mathcal{U}}\hat{\mathcal{U}}^{\top})\|$ using Eqs. (8.29), (8.30) and (8.36). The bound for $\|\mathcal{D}\|$ and $\|\mathcal{U}\|_{2\to\infty}$ follows from Assumption 3 and Assumption 4, respectively.

Combining Eqs. (8.37) and (8.38) we obtain

$$\|(\hat{\mathcal{U}}\hat{\mathcal{D}}\hat{\mathcal{U}}^{\top} - \mathcal{U}\mathcal{D}\mathcal{U}^{\top})\delta\|_{\infty} = \mathcal{O}\Big(\frac{\ln p}{n\sqrt{p}}\Big) \tag{8.39}$$

with probability at least $1 - \mathcal{O}(p^{-2})$.

Table 15: Asymptotic Order of Each Term:

| Expression | $\mathbf{v} = \delta$ | $\mathbf{v} = \Sigma^{1/2}\zeta$ | Corresponding Terms |
|---|---|---|---|
| $\|\mathcal{W}\mathbf{v}\|_\infty$ | $\sqrt{\frac{\ln p}{n}}$ | n.a. | (A) |
| $\left\|[\hat{\mathcal{U}}\hat{\mathcal{D}}\hat{\mathcal{U}}^\top - \mathcal{U}\mathcal{D}\mathcal{U}^\top]\mathbf{v}\right\|_\infty$ | $\frac{\ln p}{n\sqrt{p}}$ | $\sqrt{\frac{\ln p}{np}}\|\zeta\|$ | (B)-$(I)$ and (C)-$(I)$ |
| $\left\|(\hat{\sigma}^{-1} - \sigma^{-1})(\mathcal{I}_p - \mathcal{U}\mathcal{U}^\top)\mathbf{v}\right\|_\infty$ | $\frac{\ln p}{n}$ | $\sqrt{\frac{\ln p}{n}}\|\zeta\|_\infty$ | (B)-$(II)$ and (C)-$(II)$ |
| $\|\hat{\sigma}^{-1}(\hat{\mathcal{U}}\hat{\mathcal{U}}^\top - \mathcal{U}\mathcal{U}^\top)\mathbf{v}\|_\infty$ | $\frac{\ln p}{n}$ | $\sqrt{\frac{\ln p}{n}}\|\zeta\|$ | (B)-$(III)$ and (C)-$(III)$ |

Using similar arguments as that for Eqs. (8.37) and (8.38) we also have

$$\frac{1}{\hat{\sigma}}\|(\hat{\mathcal{U}}\hat{\mathcal{U}}^\top - \mathcal{U}\mathcal{U}^\top)\delta\|_\infty \le \frac{\|((\hat{\mathcal{U}} - \mathcal{U}\Xi) - \mathcal{U}(\mathcal{U}^\top\hat{\mathcal{U}} - \Xi))\hat{\mathcal{U}}^\top\delta\|_\infty + \|\mathcal{U}\mathcal{U}^\top T\delta\|_\infty}{\hat{\sigma}}$$

$$\le \mathcal{C}\Big(\sqrt{\frac{d^3\ln p}{np}} + \frac{\sqrt{d}\ln p}{n\sqrt{p}}\Big) \times \sqrt{\frac{p\ln p}{n}} + \mathcal{C}\sqrt{\frac{d\ln p}{np}} \times \sqrt{\frac{p\ln p}{n}} \qquad (8.40)$$

$$= \mathcal{O}\Big(\frac{\ln p}{n}\Big)$$

with probability at least $1 - \mathcal{O}(p^{-2})$, where once again $T = \mathcal{I}_p - \hat{\mathcal{U}}\hat{\mathcal{U}}^\top$. Note that in the above derivations we have used Lemma 1 to show that $\hat{\sigma}^{-1}$ is bounded away from 0 by some constant not depending on $p$ and $n$.

Finally we can also replace $\delta$ with $\Sigma^{1/2}\zeta$ in the derivations of Eqs. (8.37), (8.38) and (8.40) to obtain

$$\|(\hat{\mathcal{U}}\hat{\mathcal{D}}\hat{\mathcal{U}}^\top - \mathcal{U}\mathcal{D}\mathcal{U}^\top)\Sigma^{1/2}\zeta\|_\infty = \mathcal{O}\Big(\sqrt{\frac{\ln p}{np}}\|\zeta\|_2\Big) \qquad (8.41)$$

$$\|\hat{\sigma}^{-1}(\hat{\mathcal{U}}\hat{\mathcal{U}}^\top - \mathcal{U}\mathcal{U}^\top)\Sigma^{1/2}\zeta\|_\infty = \mathcal{O}\Big(\sqrt{\frac{\ln p}{n}}\|\zeta\|_2\Big) \qquad (8.42)$$

simultaneously, with probability at least $1 - \mathcal{O}(p^{-2})$. A summary of the bounds for the terms $(A)$, $(B)$-$(I)$ through $(B)$-$(III)$, and $(C)$-$(I)$ through $(C)$-$(III)$, are provided in Table 15. Combining the terms in this table we obtain the bound for $\|\hat{\zeta} - \zeta\|_\infty$ given in Eq. (3.5) (note that both $\|\zeta\|$ and $\|\zeta\|_\infty$ are bounded, see Assumption 2). This concludes the proof of Theorem 2.

Finally, for ease of reference, we state two collaries summarizing the main derivations in the proof of Theorem 2. These corollaries will be used in the proof of Theorem 3 below.

**Corollary 1.** *Suppose that Assumption 1 through Assumption 4 are satisfied. Let* $\mathbf{v}$ *be either a* fixed *vector in* $\mathbb{R}^p$ *or a* $p$-*variate sub-Gaussian random vector with* $\mathbb{E}[\mathbf{v}] = \mathbf{0}$. *We then have*

$$\left\|(\hat{\mathcal{W}} - \mathcal{W})\mathbf{v}\right\|_\infty = \begin{cases} \mathcal{O}\big(n^{-1/2}(\ln p)\max_i \varsigma_i\big) & \text{if } \mathbf{v} \text{ is a sub-Gaussian vector} \\ \mathcal{O}\big(n^{-1/2}(\ln p)^{1/2}\|\Sigma^{-1/2}\mathbf{v}\|\big) & \text{if } \mathbf{v} \text{ is a constant vector} \end{cases}$$

*with probability at least* $1 - \mathcal{O}(p^{-2})$, *where* $\varsigma_i^2$ *is the variance of the* $i^{th}$ *element of* $\mathbf{v}$.

**Corollary 2.** *Suppose that Assumption 1 through Assumption 4 are satisfied. Let* $\mathbf{v}$ *be a* $p$-*variate sub-Gaussian random vector with* $\text{Var}[\mathbf{v}] = c\,n^{-1}\Sigma$ *for some finite* $c > 0$. *We then have*

$$\left\|\hat{\mathcal{W}}\mathbf{v} - \mathcal{W}\mathbb{E}[\mathbf{v}]\right\|_\infty = \mathcal{O}\Big(\sqrt{n^{-1}\ln p}\Big)$$

*with probability at least* $1 - \mathcal{O}(p^{-2})$.

### 8.4 Proof of Theorem 3

For simplicity of notation we will write $\mathcal{S}$ instead of $\mathcal{S}_\zeta$ since Theorem 3 only depends on the whitened vector $\zeta$. Now recall the definition of $\tilde{\mathcal{S}}$ as

$$\tilde{\mathcal{S}} = \{j : |\hat{\zeta}_j| > t_n\}$$

where $t_n = \left(\frac{\ln p}{n}\right)^\alpha$ for some constant $0 < \alpha < \frac{1}{2}$. We now show $\tilde{\mathcal{S}} = \mathcal{S}$ asymptotically almost surely.

From Theorem 2 there exists a choice of $\mathcal{C}$ such that if $\beta_n = \mathcal{C}\sqrt{n^{-1}\ln p}$ then

$$\mathbb{P}\Big(\bigcup_{j=1}^p \{|\zeta_j - \hat{\zeta}_j| > \beta_n\}\Big) = \mathcal{O}(p^{-2}).$$

Now suppose $\mathcal{S}^c \cap \tilde{\mathcal{S}} \neq \emptyset$ where $(\cdot)^c$ denote set complement. Then there exists a $j$ such that $\zeta_j = 0$ and $|\hat{\zeta}_j| > t_n$, and for this $j$ we have $|\zeta_j - \hat{\zeta}_j| > t_n > \beta_n$, provided that $n$ is sufficiently large. We thus have

$$\mathbb{P}(\mathcal{S}^c \cap \tilde{\mathcal{S}} \neq \emptyset) \leq \mathbb{P}\Big(\bigcup_{j=1}^p \{|\zeta_j - \hat{\zeta}_j| > \beta_n\}\Big) = \mathcal{O}(p^{-2}). \tag{8.43}$$

Similarly, if $\mathcal{S} \cap \tilde{\mathcal{S}}^c \neq \emptyset$ then there exist a $j$ such that $|\zeta_j| > \mathcal{C}_0$ and $|\hat{\zeta}_j| \leq t_n$. Recall that $\mathcal{C}_0 > 0$ is the constant appearing in Assumption 2; in particular, $\mathcal{C}_0$ does not depend on $n$ and $p$. By the reverse triangle inequality, $|\zeta_j - \hat{\zeta}_j| > \mathcal{C}_0 - t_n > \beta_n$ for sufficiently large $n$ and hence

$$\mathbb{P}(\mathcal{S} \cap \tilde{\mathcal{S}}^c \neq \emptyset) \leq \mathbb{P}\Big(\bigcup_{j=1}^p \{|\zeta_j - \hat{\zeta}_j| > \beta_n\}\Big) = \mathcal{O}(p^{-2}). \tag{8.44}$$

Combining Eq. (8.43) and Eq. (8.44) yields $\mathbb{P}(\mathcal{S} \neq \tilde{\mathcal{S}}) = \mathcal{O}(p^{-2})$ and hence, as $p \to \infty$, by the Borel-Cantelli lemma we have $\mathcal{S} = \tilde{\mathcal{S}}$ asymptotically almost surely.

We now show that the error rate for lda $\circ$ pca converges to the Bayes error rate $R_F$ asymptotically almost surely. From the description of lda $\circ$ pca in Eq. (2.9), it is sufficient to show that

$$\hat{\zeta}_{\tilde{\mathcal{S}}}^\top \big[\hat{\mathcal{W}}\big(\mathbf{Z} - \tfrac{\bar{X}_1 + \bar{X}_2}{2}\big)\big]_{\tilde{\mathcal{S}}} - \zeta_{\mathcal{S}}^\top \big[\mathcal{W}\big(\mathbf{Z} - \tfrac{\mu_1 + \mu_2}{2}\big)\big]_{\mathcal{S}} \xrightarrow{\mathrm{P}} 0, \tag{8.45}$$

$$\ln \frac{n_1}{n_2} \xrightarrow{\mathrm{P}} \ln \frac{\pi_1}{1 - \pi_1}. \tag{8.46}$$

Note that the convergence in Eq. (8.45) is with respect to a random testing sample $\mathbf{Z} \sim \pi_1 \mathcal{N}(\mu_1, \Sigma) + (1 - \pi_1)\mathcal{N}(\mu_2, \Sigma)$ together with the training data, while the convergence in Eq.equation 8.46 is with respect to the training data only. As Eq. equation 8.46 follows directly from the strong law of large numbers, we thus focus our efforts on showing Eq. (8.45).

First, suppose $\tilde{\mathcal{S}} = \mathcal{S}$ and let $h(\mathbf{Z}) = \hat{\zeta}_{\tilde{\mathcal{S}}}^\top \big[\hat{\mathcal{W}}\big(\mathbf{Z} - \tfrac{\bar{X}_1 + \bar{X}_2}{2}\big)\big]_{\tilde{\mathcal{S}}} - \zeta_{\mathcal{S}}^\top \big[\mathcal{W}\big(\mathbf{Z} - \tfrac{\mu_1 + \mu_2}{2}\big)\big]_{\mathcal{S}}$. Then

$$\begin{aligned}
|h(\mathbf{Z})| \leq{}& s_0 \big\|\big[\hat{\mathcal{W}}\big(\mathbf{Z} - \tfrac{\bar{X}_1 + \bar{X}_2}{2}\big) - \mathcal{W}\big(\mathbf{Z} - \tfrac{\mu_1 + \mu_2}{2}\big)\big]_{\mathcal{S}}\big\|_\infty \|\zeta\|_\infty \\
&+ s_0 \big\|\big[\mathcal{W}\big(\mathbf{Z} - \tfrac{\mu_1 + \mu_2}{2}\big)\big]_{\mathcal{S}}\big\|_\infty \|\hat{\zeta} - \zeta\|_\infty \\
&+ s_0 \big\|\big[\hat{\mathcal{W}}\big(\mathbf{Z} - \tfrac{\bar{X}_1 + \bar{X}_2}{2}\big) - \mathcal{W}\big(\mathbf{Z} - \tfrac{\mu_1 + \mu_2}{2}\big)\big]_{\mathcal{S}}\big\|_\infty \|\hat{\zeta} - \zeta\|_\infty.
\end{aligned} \tag{8.47}$$

The bounds for $\|\zeta\|_\infty$ and $\|\hat{\zeta} - \zeta\|_\infty$ are given in Assumption 2 and Theorem 2, respectively. It thus suffices to bound

$$(D) := \big\|\big[\hat{\mathcal{W}}\big(\mathbf{Z} - \tfrac{\bar{X}_1 + \bar{X}_2}{2}\big) - \mathcal{W}\big(\mathbf{Z} - \tfrac{\mu_1 + \mu_2}{2}\big)\big]_{\mathcal{S}}\big\|_\infty$$

$$(E) := \big\|\big[\mathcal{W}\big(\mathbf{Z} - \tfrac{\mu_1 + \mu_2}{2}\big)\big]_{\mathcal{S}}\big\|_\infty$$

Write $\mathbf{Z} = \mu_z + \Sigma^{1/2}\epsilon_z$ where $\mu_z = \mu_1$ if $\mathbf{Z}$ is sampled from class 1 and $\mu_z = \mu_2$ otherwise. The term $\big\|[(\hat{\mathcal{W}} - \mathcal{W})\Sigma^{1/2}\epsilon_z]_{\mathcal{S}}\big\|_\infty$ can be analyzed using the decomposition for $\hat{\mathcal{W}} - \mathcal{W}$ given in Eq. (8.25) together

with similar arguments to that for deriving Eqs. (8.27), (8.37), (8.38) and (8.40). In particular we have, with probability at least $1 - \mathcal{O}(n^{-2})$, that

$$\left\| \left[ (\hat{\mathcal{W}} - \mathcal{W}) \Sigma^{1/2} \epsilon_z \right]_{\mathcal{S}} \right\|_\infty = \mathcal{O}\Big(\sqrt{\frac{\ln n \ln p}{n}}\Big) \tag{8.48}$$

Next, using Corollary 1, Corollary 2 and Eq. (8.48), we obtain, with probability at least $1 - \mathcal{O}(n^{-2})$

$$(D) \leq \left\| \left[ (\hat{\mathcal{W}} - \mathcal{W}) \Sigma^{1/2} \epsilon_z \right]_{\mathcal{S}} \right\|_\infty + \left\| (\hat{\mathcal{W}} - \mathcal{W}) \mu_z \right\|_\infty + \frac{1}{2} \left\| \hat{\mathcal{W}} \bar{X}_1 - \mathcal{W} \mu_1 \right\|_\infty + \frac{1}{2} \left\| \hat{\mathcal{W}} \bar{X}_2 - \mathcal{W} \mu_2 \right\|_\infty$$

$$\leq \mathcal{C}\Big( \sqrt{\frac{\ln n \ln p}{n}} + \sqrt{\frac{\ln p}{n}} + \sqrt{\frac{\ln p}{n}} \Big) = \mathcal{O}\Big(\sqrt{\frac{\ln n \ln p}{n}}\Big).$$

Thirdly, we have

$$(E) \leq \left\| \mathcal{W}\big(\mu_z - \tfrac{\mu_1 + \mu_2}{2}\big) \right\|_\infty + \left\| \left[ \mathcal{W}(\mathbf{Z} - \mu_z) \right]_{\mathcal{S}} \right\|_\infty = \frac{1}{2} \left\| \zeta \right\|_\infty + \left\| [\epsilon_z]_{\mathcal{S}} \right\|_\infty =: \vartheta(\mathbf{Z}) \tag{8.49}$$

where $[\epsilon_z]_{\mathcal{S}}$ is a mean 0 sub-Gaussian vector in $\mathbb{R}^{s_0}$ and $\mathrm{Var}[[\epsilon_z]_{\mathcal{S}}] = \mathcal{I}_{s_0}$. Therefore, by Assumption 2 and properties of sub-Gaussian random vectors, the term $\vartheta(\mathbf{Z})$ is bounded in probability.

Combining the above bounds, we conclude that with probability at least $1 - \mathcal{O}(n^{-2})$, $\tilde{\mathcal{S}} = \mathcal{S}$ and

$$\left| \hat{\zeta}_{\tilde{\mathcal{S}}}^\top \big[ \hat{\mathcal{W}}(\mathbf{Z} - \tfrac{\bar{X}_1 + \bar{X}_2}{2}) \big]_{\tilde{\mathcal{S}}} - \zeta_{\mathcal{S}}^\top \big[ \mathcal{W}(\mathbf{Z} - \tfrac{\mu_1 + \mu_2}{2}) \big]_{\mathcal{S}} \right| \leq \mathcal{C} s_0 \Big( \sqrt{\frac{\ln n \ln p}{n}} \left\| \zeta \right\|_\infty + \vartheta(\mathbf{Z}) \sqrt{\frac{\ln p}{n}} \Big).$$

Hence, for $\ln n \ln p = o(n)$, we have

$$\left| \hat{\zeta}_{\tilde{\mathcal{S}}}^\top \big[ \hat{\mathcal{W}}(\mathbf{Z} - \tfrac{\bar{X}_1 + \bar{X}_2}{2}) \big]_{\tilde{\mathcal{S}}} - \zeta_{\mathcal{S}}^\top \big[ \mathcal{W}(\mathbf{Z} - \tfrac{\mu_1 + \mu_2}{2}) \big]_{\mathcal{S}} \right| \longrightarrow 0$$

in probability. This completes the proof of Theorem 3.

## 8.5 Theoretical results for Section 5.1

We now present theoretical results for multi-class lda ∘ pca. We first assume that the whitened directions $\{\zeta^{(i)}\}$, the whitened indices $\{\mathcal{S}_i\}$, and the covariance $\Sigma$ satisfy the following two conditions, which are natural generalizations of Assumption 2 and Assumption 3 to the multi-class setting.

*Assumption 6.* Let $|\mathcal{S}_i| = s_i > 0$ for each $i = 2, \cdots, K$. Recall that $\mathcal{S}_i$ is the set of indices $j$ for which $\zeta_{ij} \neq 0$. Let $\mathcal{C}_0 > 0$, $M > 0$ and $\mathcal{C}_\zeta > 0$ be constants not depending on $p$ such that $\max_i s_i \leq M$ and

$$\min_{i=2,\cdots,K} \min_{j \in \mathcal{S}_i} |\zeta_{ij}| \geq \mathcal{C}_0, \qquad \max_{i \in [K]} \|\Sigma^{-1/2} \mu_i\| \leq \mathcal{C}_\zeta.$$

*Assumption 7.* Let $\sigma > 0$ be fixed and that, for sufficiently large $p$, $n_1, \cdots, n_K$ and $p$ satisfy

$$\frac{n_i}{n_j} = \Theta(1) \text{ for } i \neq j, \ i, j \in [K] \quad \text{and} \quad \ln p = o(n).$$

Furthermore, for sufficiently large $p$, the spiked eigenvalues $\lambda_1, \ldots, \lambda_d$ satisfy

$$\lambda_k = \Theta(p), \qquad \text{for all } k \in [d].$$

Given the above conditions, the next result extends Theorem 2 (and has an identical proof) to bound $\|\hat{\zeta}^{(i)} - \zeta^{(i)}\|_\infty$ for $i \geq 2$.

**Theorem 4.** *Under Assumption 1, Assumption 4, Assumption 6 and Assumption 7, there exists a constant $C > 0$ such that with probability at least $1 - \mathcal{O}(p^{-2})$,*

$$\max_{i=2,\cdots,K} \|\hat{\zeta}^{(i)} - \zeta^{(i)}\|_\infty \leq \mathcal{C}\sqrt{\frac{\ln p}{n}}. \tag{8.50}$$

We now consider a hard thresholding estimate $\tilde{\zeta}^{(i)}$ for recovering $\zeta^{(i)}$. For a given $i \geq 2$, let

$$\tilde{\zeta}_j^{(i)} = \hat{\zeta}_j^{(i)} \mathbb{1}(|\hat{\zeta}_j^{(i)}| > t_n), \qquad j \in [p] \tag{8.51}$$

where $t_n = \left(\ln p/n\right)^\alpha$ for some constant $0 < \alpha < \frac{1}{2}$. Here we assume, for simplicity, that $\alpha$ takes the same value for all classes. Given $\tilde{\zeta}^{(i)}$, define the associated active set $\tilde{\mathcal{S}}_i = \{j : \tilde{\zeta}_j^{(i)} \neq 0\}$. The next result extends Theorem 3 (and has an identical proof) to show that lda $\circ$ pca is also asymptotically Bayes-optimal in the multi-class setting. However, we note that (to the best of our knowledge), there is no closed-form explicit expression for the Bayes error $R_F$ when classifying data from a mixture of $K \geq 3$ multiviarate normals.

**Theorem 5.** *Suppose that* $\mathbf{Z} \sim \sum_{i=1}^K \pi_i \mathcal{N}_p(\mu_i, \Sigma)$ *where* $\pi_i \geq 0$ *and* $\sum_{i=1}^K \pi_i = 1$. *Suppose Assumption 1, Assumption 4, Assumption 6 and Assumption 7 are satisfied. We then have*

$$\max_{i=2,\cdots,K} \mathbb{P}(\tilde{\mathcal{S}}_i \neq \mathcal{S}_i) = \mathcal{O}(p^{-2}). \tag{8.52}$$

*Furthermore we also have* $\hat{R}_{\text{lda}\circ\text{pca}} - R_F \to 0$ *almost surely as* $n, p \to \infty$.

## 8.6 Theoretical results for Section 5.2

We now present theoretical results for qda $\circ$ pca. which depend on the following variant of Assumption 2 through Assumption 4 for heterogeneous covariance matrices.

*Assumption 8.* Let $|\mathcal{A}_i| = a_i > 0$. Recall that $\mathcal{A}_i$ is the set of indices $j$ for which $\zeta_{ij} \neq 0$, $i = 1, 2$ where $\zeta_i = \Sigma_i^{-1/2}\mu_i$. Let $\mathcal{C}_0 > 0$, $M > 0$ and $\mathcal{C}_\zeta > 0$ be constants not depending on $p$ such that $\max\{a_1, a_2\} \leq M$ and

$$\min_{i\in\{1,2\}} \min_{j\in\mathcal{A}_i} |\zeta_{ij}| \geq \mathcal{C}_0, \qquad \max_{i\in\{1,2\}} \max_{k\in\{1,2\}} \|\Sigma_i^{-1/2}\mu_k\| \leq \mathcal{C}_\zeta.$$

*Assumption 9.* Let $\sigma_i > 0$ be fixed and that, for sufficiently large $p$, $n_1, n_2$ and $p$ satisfy

$$\frac{n_1}{n_2} = \Theta(1), \quad \ln p = o(n).$$

Furthermore, for sufficiently large $p$, the spiked eigenvalues $\lambda_{i1}, \ldots, \lambda_{id_i}$ satisfy

$$\lambda_{ik} = \mathcal{O}(p), \qquad \text{for all } k \in [d_i], \, i = 1, 2.$$

*Assumption 10* (Bounded Coherence). There is a constant $\mathcal{C}_\mathcal{U} \geq 1$ independent of $n$ and $p$ such that

$$\|\mathcal{U}_i\|_{2\to\infty} \leq \frac{\mathcal{C}_\mathcal{U}\sqrt{d_i}}{\sqrt{p}}, \qquad \text{for } i = 1, 2.$$

Assumption 8 guarantees that the noncentrality parameters for these $\chi_1^2$ are strictly positive and finite, so that the Bayes error rate $R_F$ is strictly bounded away from 0 and $\min\{\pi_1, 1 - \pi_1\}$ (which corresponds to random guessing). Note that if $\Sigma_1 \neq \Sigma_2$ then there is no simple closed-form expression for $R_F$ as it depends on the tail behavior of a linear combination of independent, *noncentral* $\chi_1^2$ random variables; see Anderson (2003, Section 6.10) for more details. Assumption 9 allows the spiked eigenvalues for each $\Sigma_i$ to grow linearly with the dimension $p$, in contrast to the bounded eigenvalues assumption frequently encountered in the literature (Li & Shao, 2015; Cai & Zhang, 2021).

We then have the following extensions of Theorem 2 and Theorem 3.

**Theorem 6.** *Under Assumption 5 through 10, there exists a constant* $C > 0$ *such that*

$$\max_{i=1,2} \|\hat{\zeta}_i - \zeta_i\|_\infty \leq C\sqrt{\frac{\ln p}{n}}. \tag{8.53}$$

*with probability at least* $1 - O(p^{-2})$.

*Construct $\tilde{\zeta}_i$ as in Eq. (3.6) for $i \in \{1, 2\}$ and let $\tilde{\mathcal{A}}_i$ be the indices for the non-zero coordinates of $\tilde{\zeta}_i$. Let $\mathbf{Z} \sim \pi_1 \mathcal{N}_p(\mu_1, \Sigma_1) + (1 - \pi_1) \mathcal{N}_p(\mu_2, \Sigma_2)$. Then*

$$\max_{i=1,2} \mathbb{P}(\tilde{\mathcal{A}}_i \neq \mathcal{A}_i) = \mathcal{O}(p^{-2}). \tag{8.54}$$

*Furthermore, we also have $\hat{R}_{\text{qdaopca}} - R_F \to 0$ in probability, as $n, p \to \infty$.*

*Proof.* For conciseness, we omit the derivations of Eq. (8.53) and Eq. (8.54) as they follow the same argument as that in the proof of Theorem 2 and Eq. (3.8).

In order to show that $Q(\mathbf{Z} \mid \bar{X}_i, \hat{\mathcal{W}}_i, \hat{\mathcal{A}}_0)$ is a consistent estimate for $Q(\mathbf{Z} \mid \mu_i, \mathcal{W}_i, \mathcal{A}_0)$, it suffices to show that the following quantities

$$q_1(\mathbf{Z}) := \left| \left[ \mathcal{W}_1(\mathbf{Z} - \mu_1) \right]_{\mathcal{A}_0}^{\top} \left[ \mathcal{W}_1(\mathbf{Z} - \mu_1) \right]_{\mathcal{A}_0} - \left[ \hat{\mathcal{W}}_1(\mathbf{Z} - \bar{X}_1) \right]_{\mathcal{A}_0}^{\top} \left[ \hat{\mathcal{W}}_1(\mathbf{Z} - \bar{X}_1) \right]_{\mathcal{A}_0} \right| \tag{8.55}$$

$$q_2(\mathbf{Z}) := \left| \left[ \mathcal{W}_2(\mathbf{Z} - \mu_2) \right]_{\mathcal{A}_0}^{\top} \left[ \mathcal{W}_2(\mathbf{Z} - \mu_2) \right]_{\mathcal{A}_0} - \left[ \hat{\mathcal{W}}_2(\mathbf{Z} - \bar{X}_2) \right]_{\mathcal{A}_0}^{\top} \left[ \hat{\mathcal{W}}_2(\mathbf{Z} - \bar{X}_2) \right]_{\mathcal{A}_0} \right| \tag{8.56}$$

both converge to 0 as $n \to \infty$.

Assume without loss of generality that $\mathbf{Z}$ is a test sample from class 1, i.e,. $\mathbf{Z} \sim \mathcal{N}_p(\mu_1, \Sigma_1)$. Then

$$\begin{aligned} q_1(\mathbf{Z}) &\leq a_0 \left\| [\mathcal{W}_1(\mathbf{Z} - \mu_1) - \hat{\mathcal{W}}_1(\mathbf{Z} - \bar{X}_1)]_{\mathcal{A}_0} \right\|_\infty^2 \\ &+ 2a_0 \left\| \mathcal{W}_1(\mathbf{Z} - \mu_1)]_{\mathcal{A}_0} \right\|_\infty \times \left\| [\mathcal{W}_1(\mathbf{Z} - \mu_1) - \hat{\mathcal{W}}_1(\mathbf{Z} - \bar{X}_1)]_{\mathcal{A}_0} \right\|_\infty \end{aligned} \tag{8.57}$$

where $a_0 = |\mathcal{A}_0| \leq a_1 + a_2$, with $a_i = |\mathcal{A}_i|$.

Let $h_1(\mathbf{Z}) := \left\| [\mathcal{W}_1(\mathbf{Z} - \mu_1) - \hat{\mathcal{W}}_1(\mathbf{Z} - \bar{X}_1)]_{\mathcal{A}_0} \right\|_\infty$. Then by Corollaries 1 and 2, and following the same arguments as that for Eq. (8.48), we have

$$\begin{aligned} h_1(\mathbf{Z}) &\leq \|(\mathcal{W}_1 - \hat{\mathcal{W}}_1)\mu_1\|_\infty + \|[(\mathcal{W}_1 - \hat{\mathcal{W}}_1)\Sigma_1^{1/2}\epsilon_z]_{\mathcal{A}_0}\|_\infty + \|\hat{\mathcal{W}}_1 \bar{X}_1 - \mathcal{W}_1 \mu_1\|_\infty \\ &\leq \mathcal{C}\left( \sqrt{\frac{\ln p}{n}} + \sqrt{\frac{\ln n \ln p}{n}} + \sqrt{\frac{\ln p}{n}} \right) \end{aligned} \tag{8.58}$$

with probability at least $1 - \mathcal{O}(n^{-2})$, where $\epsilon_z \sim \mathcal{N}(0, \mathcal{I}_p)$. An almost identical bound also holds for $h_2(\mathbf{Z}) := \left\| [\mathcal{W}_2(\mathbf{Z} - \mu_2) - \hat{\mathcal{W}}_2(\mathbf{Z} - \bar{X}_2)]_{\mathcal{A}_0} \right\|_\infty$, namely

$$\begin{aligned} h_2(\mathbf{Z}) &\leq \|(\mathcal{W}_2 - \hat{\mathcal{W}}_2)\mu_1\|_\infty + \|[(\mathcal{W}_2 - \hat{\mathcal{W}}_2)\Sigma_1^{1/2}\epsilon_z]_{\mathcal{A}_0}\|_\infty + \|\hat{\mathcal{W}}_2 \bar{X}_2 - \mathcal{W}_2 \mu_2\|_\infty \\ &\leq \mathcal{C}\left( \sqrt{\frac{\ln p}{n}} \|\Sigma_2^{-1/2}\mu_1\|_2 + \sqrt{\frac{\ln n \ln p}{n}} + \sqrt{\frac{\ln p}{n}} \right) \end{aligned} \tag{8.59}$$

with probability at least $1 - \mathcal{O}(n^{-2})$.

Next, $\left\| \left[ \mathcal{W}_1(\mathbf{Z} - \mu_1) \right]_{\mathcal{A}_0} \right\|_\infty$ and $\left\| \left[ \mathcal{W}_1(\mathbf{Z} - \mu_1) \right]_{\mathcal{A}_0} \right\|_\infty$ can be bounded by the quantity $\vartheta(\mathbf{Z})$ as defined in Eq. (8.49) which, by Assumption 8, is bounded in probability.

Finally, by Proposition 2 and Lemma 2 in Jiang et al. (2018), we have $|\hat{\kappa} - \kappa| \to 0$ in probability. Combining the above statements yield the proof of Theorem 6. $\qquad\square$

