# OpenReview forum: "Classification of high-dimensional data with spiked covariance matrix structure"
_TMLR — Accepted by TMLR_

### Review · Reviewer_2mqw · 2025-08-07

**Summary Of Contributions:**

This paper introduces and analyzes a classifier named lda$\cdot$pca for high-dimensional data, where the number of features p can be much larger than the number of observations n. The core idea is to first use Principal Component Analysis (PCA) for dimension reduction and data whitening, followed by applying Linear Discriminant Analysis (LDA) in the reduced and whitened space.

The algorithm itself is natural and straightforward, while the main contribution of this paper lies in the theoretical part. The authors provide a rigorous theoretical analysis, demonstrating that under certain conditions, the lda$\cdot$pca classifier is asymptotically Bayes-optimal. This means its misclassification error converges to the lowest possible error rate as the sample size increases. This is a significant result, as it is claimed to be the first of its kind for a classifier that uses PCA for dimension reduction prior to classification in this high-dimensional setting.

We also want to mention that the theoretical framework is notably built upon a spiked covariance model where the large "spike" eigenvalues can grow with the dimension p.

**Additional Comments:**

No.

**Audience:**

Yes

**Audience Explanation:**

Yes, from my perspective, the findings of this paper would be of interest to a significant portion of TMLR's audience. Here are the primary reasons:


High-Dimensional Data: The "p >> n" problem is a central challenge in modern machine learning. Any novel method that provides a practical solution with strong theoretical backing is highly relevant.

Theoretical Guarantees: The machine learning community, and TMLR's audience in particular, values not just algorithms that work well in practice, but also a clear understanding of why and when they work. The paper's main theoretical result—Bayes optimality under a realistic spiked covariance model—is a strong and appealing contribution.

Feature Selection and Interpretability: The paper's finding that lda$\cdot$pca often selects a much smaller feature set is a major selling point. In many applications, building a simpler, more interpretable model is as important as achieving high accuracy. This aspect of the work will appeal to researchers and practitioners focused on explainable AI  and model parsimony.

**Broader Impact Concerns:**

No.

**Claims And Evidence:**

Yes

**Claims Explanation:**

The claims made in the submission are, for the most part, supported by accurate, convincing, and clear evidence.

Claim: The lda$\cdot$pca classifier is asymptotically Bayes-optimal.
Evidence: This claim is supported by Theorem 3, which states that under Assumptions 1-4, the misclassification rate of lda$\cdot$pca converges to the Bayes error rate.

Theorem 1 provides a crucial bound on the estimation error of the eigenvectors of the covariance matrix, and Theorem 2 bounds the estimation error of the whitened direction ζ. These intermediate results build the foundation for the main theorem. The authors also clearly distinguish their theoretical assumptions (e.g., diverging eigenvalues) from prior work, strengthening the novelty and relevance of their result. The evidence is convincing from my perspective.

Claim: lda$\cdot$pca is competitive with state-of-the-art methods in terms of accuracy.
Evidence: This is supported by the numerical results in Tables 1, 3, 5, 7, and 8.

Claim: lda$\cdot$pca selects a smaller number of features.
Evidence: This is supported by the model size comparisons in Tables 2, 4, 6, 7, and 8.

**Requested Changes:**

The paper is well-written and the contributions are significant. Following are some of my personal thoughts:

1. The classifier has two main tuning parameters: d (the number of spikes) and s (the sparsity level of the whitened direction). For d, the paper suggests choosing it to account for 90% of the variability. For s, it is selected via cross-validation. While the paper mentions that
d can be consistently estimated, a more detailed discussion or sensitivity analysis regarding the choice of the 90% threshold for
d would be beneficial. How sensitive is the final performance to this choice? A brief experiment showing the impact of varying this threshold could add practical value.

2. Computational Complexity: The paper does not include a discussion on the computational complexity of the lda$\cdot$pca algorithm compared to the other methods. Adding a brief section or paragraph on this would be helpful for practitioners who need to decide which method to use on very large datasets. The PCA step is **generally efficient**, but a formal comparison would be a valuable addition. (There is no need to add computational complexity comparison in the review round, just my curiosity.)


3. Performance on Dense Signal Models: In the simulation studies for Model 1 (equal correlation) and Model 3 (random correlation), the true whitened direction ζ is dense (all entries are non-zero). The paper notes that lda$\cdot$pca works well by capturing the few "strong" signals and ignoring the many "weak" ones. While this is a strength, it would be interesting to see how it performs in a scenario with a truly sparse ζ that has only a few non-zero entries of similar magnitude. This would be a direct test of the feature selection mechanism in a less ambiguous setting and would complement the existing Model 2 (block diagonal), which is sparse in both the original and whitened domains.

---

> ### Author Response · Authors · 2025-10-13
> **Response to Reviewer 2mqw for Paper 5423**
>
> For (1): A related sensitivity analysis is presented in Section~7.2, where performance is evaluated under varying $d=1,2,3$ in a spiked covariance model with two true spikes. Our findings show that underestimating $d$ significantly degrades performance while overestimating $d$ has minimal impact on classification error. Even when $d$ is overestimated, our method selects a number of features ($3.82$ on average) close to the true sparsity level ($3$), compared to $49.35$ selected by DSDA. We advise choosing $d$ in practice as the smallest integer satisfying the $90\%$ cumulative eigenvalue threshold.
>
> For (2): Thank you for your helpful suggestion. We agree that computational complexity is important for practical applications. To address this, we have included a complexity analysis in Section 2.4.2. As discussed there, our method is particularly efficient in settings with low-rank or latent variable structures, where PCA-based dimension reduction provides both statistical and computational benefits.
>
> For (3): We have added a corresponding sensitivity analysis for varying choices of $d$ in Section~7.2, where we ensure that  $\mu_{2}-\mu_{1}$, $\beta$ and $\zeta$ share the same sparsity level, located at the first three coordinates.

---

> > ### Author Response · Authors · 2025-11-25
> > **Follow up with Reviewer 2mqw**
> >
> > We appreciate the feedback and comments from reviewer 2mqw. Please let us know if there are still outstanding questions and concerns, thank you.

---

### Review · Reviewer_eQPd · 2025-08-18

**Summary Of Contributions:**

The authors study classficiation using the PCA + LDA approach, wherein first a PCA is used to decrease the dimension of the data and then LDA is applied for classification. They show theoretically that this procedure reaches the Bayes error asymptotically under mild assumptions, in particular only assuming that log(p)/n  goes to zero. They also extend their results beyong Gaussianity and two classes.

**Audience:**

Yes

**Audience Explanation:**

I think these are genuinely interesting results for a broad audience in statistics and machine learning. I was not even aware that PCA + LDA are used together and the theoretical results the authors present give a strong argument for this procedure.

**Claims And Evidence:**

Yes

**Claims Explanation:**

I am not an expert on the underlying theory and literature, but the results and proofs, as well as the empirical results, appear solid. The whole paper including the proofs are well-written and the reasoning clearly explained.

**Requested Changes:**

This paper was a pleasure to read; well-written, well explained and with relevant results and an impressive literature overview. The proofs are well-written as well and easy to follow, with several smart writing choices that help the flow of reading, such as collecting often used algebraic facts in one place.

My only real "concern" is the length of the paper, specifically Section 5, which not only makes the paper longer but also disturbs the flow of the story. The way I understand the contribution, even without Section 5 the paper would be interesting enough for TMLR, so there is the question of whether this section is really needed. On the other hand, I see that that might not be enough for a separate paper, and it may also not be ideal to move it into the appendix.

Aside from that I only have minor comments/requests:
- What exactly are the assumptions of d and s relativ to p? In the experimental section, Model 1 has no sparsity in the withened data, i.e. s=p. Is this a problem from a theoretical point of view or is this allowed? If so, would the method correctly identify p=s as the sample size increases? This may be spelled out more clearly in the main text.
- This is a detail but it may be worth writing that Step 1 on page 5 is the same as PCA, depending on the audience this might not be immediately clear.
- Page 11, when talking about Sifaou et al relying heavy on random matrix theory, maybe it could makes sense to very briefly spell out again what you do differently to allow for $p/n \to \infty$.
- The Bernstein Inequality for random vectors is used several times in the proofs. It might be nice to quickly add it somewhere in the beginning of the proof section.


- page 2 first paragraph, "features selection" --> "feature selection"

---

> ### Author Response · Authors · 2025-10-13
> **Response to Reviewer eQPd for Paper 5423**
>
> We appreciate the feedback and to address the concern, we have moved the section on elliptical distributions to the appendix. We agree that the core contribution of the paper stands on its own. However, we believe Section 5 adds meaningful value by addressing important gaps in the existing literature. Specifically, while previous works rely on sophisticated assumptions to extend to multi-class and QDA settings, Section 5 highlights the applicability of our theoretical framework under a similar spiked covariance structure.
>
> For (1): We have addressed this concern in Assumption 2, Remark 1, and Remark 6. Briefly, our theoretical analysis assumes fixed $d$ and $s_{0}$. Models 1 and 3 are deliberately designed to examine the behavior of our method beyond the sparsity regime where $s_{0}=p$. While full recovery of $\beta$ or $\zeta$ is not expected in such settings (see further discussion in the second paragraph of page 2), $\mathrm{lda} \circ \mathrm{pca}$ still delivers competitive performance.
>
> For (2): Thank you for the suggestion. We have clarified that Step 1 corresponds exactly to PCA in the descriptions of the standard $\mathrm{lda} \circ \mathrm{pca}$, $K$-classes $\mathrm{lda} \circ \mathrm{pca}$ and $\mathrm{qda} \circ \mathrm{pca}$ decision rules.
>
> For (3):  Thank you for the comment. Our work differs by (1) imposing sparsity assumptions to control estimation error motivated by earlier research (see more details in the Introduction) and (2) leveraging sharp entrywise perturbation bounds for precise eigenspace estimation. We have added a brief discussion of these differences at the end of the Remark~4 for clarity.
>
> For (4): Thank you for the helpful suggestion. For completeness, we have added a statement of Bernstein’s inequality in the Appendix to provide a clear reference for the reader.
>
> For (5): Thank you for pointing this out. We corrected “features selection” to “feature selection” in the revised version.

---

> > ### Comment · Reviewer_eQPd · 2025-11-01
> > **My comments have been addressed**
> >
> > I feel my questions/remarks have been adequately addressed, thank you !

---

### Review · Reviewer_fxez · 2025-11-24

**Summary Of Contributions:**

Please disregard this as my review was mistakenly made as an official comment, and my questions have been addressed.

**Additional Comments:**

Please disregard this as my review was mistakenly made as an official comment, and my questions have been addressed.

**Audience:**

Yes

**Audience Explanation:**

Please disregard this as my review was mistakenly made as an official comment, and my questions have been addressed.

**Broader Impact Concerns:**

Please disregard this as my review was mistakenly made as an official comment, and my questions have been addressed.

**Claims And Evidence:**

Yes

**Claims Explanation:**

Please disregard this as my review was mistakenly made as an official comment, and my questions have been addressed.

**Requested Changes:**

Please disregard this as my review was mistakenly made as an official comment, and my questions have been addressed.

---

### Comment · Reviewer_fxez · 2025-09-07
**Review of TMLR Paper5423**

## Summary of the work
- This paper considers the problem of High-dimensional binary classification with spiked-structure covariance matrix and sparse mean difference
- The proposed method uses PCA to estimate a whitening transform. The whitened mean difference $\hat{\zeta}$ is computed and the  \(s\) largest elements are kept. Fisher LDA is run on the selected, whitened features, where d (spikes/factors) and s (features kept).
- Theoretically, the authors show that the leading eigenvectors are well estimated, and have quantified the entry-wise accuracy and show that the method recovers the true support and achieves Bayes-rate error.

## Strengths
- This work is clear, well structured, and easy to follow. The algorithmic steps are explicit, the assumptions are well stated; though the practical guidance for choosing d and s  is somewhat scattered.
- Minor typos (e.g., “emprical”, “dateset”); easy to fix.
- The paper is conceptually simple and intuitive to understand. The authors have taken advantage of the principled link between PCA and LDA.
- I understand that the novelty of the work is somewhat twofold; the work avoids the advantages of sparse-$\beta$ approaches, and their asymptotic regime has allowed for diverging spikes, which can be more realistic in some practical settings.

## Comments to be Addressed

- Is there a mismatch between the abstract and the main text? The abstract claims Bayes optimality “whenever $s \sqrt(\ln p/n)\rightarrow 0$,” suggesting s may grow; Section 3 effectively imposes fixed sparsity $(|S_\zeta|\leq M)$ and then requires $ \ln p=O(n)$. Can we clarify the growth allowed for s and give the exact joint scaling?
- Selecting the number of spikes d: In the Methodology, the authors claim that d is a tuning parameter that can be consistently estimated. In Section 3 they further add a concrete eigenvalue-ratio rule which recovers the true d asymptotically. Since the choice of d affects the classification risk, proper choice of d is important, What I believe is missing here is a clear, practical, reproducible rule for choosing d, as well as a demonstration of how the results can change if d is not exact, such as small sensitivity analysis (e.g., test error vs. d).
- For reproducibility, can the authors share how the sparsity  s or the threshold  \hat{\eta} is chosen in practice?
- what is the difference in computational complexity between sparse-\beta baselines and the proposed method?

---

> ### Author Response · Authors · 2025-10-13
> **Response to Reviewer fxez for Paper 5423**
>
> For (1): We have clarified the assumptions on $s$ and $d$ in the revised Assumption 2. Specifically, we assume that the number of spikes $d$ is fixed and does not grow with $p$ or $n$, and a similar assumption applies to the sparsity level $s_0 = |\mathcal{S}_\zeta|$.
>
> While $s_0$ is fixed for technical convenience, the results can be handled \emph{mutatis mutandis} with $s_0 = o\left(\left(\frac{n}{\ln p}\right)^\tau\right), \text{for some } \tau \in [0, 1/2)$, which ensures our method is Bayes optimal mentioned in the abstract when $s_{0} \sqrt{n^{-1}\ln p} \to 0 $.
>
> For (2): To address this, we have included a sensitivity analysis in Section~7.2, where we evaluate classification performance across different choices of $d$. This analysis examines a modified spiked covariance model with two true spikes and compares the cases $d=1$, $d=2$ (correct) and $d=3$. Our findings indicate that underestimating $d$ leads to a significant increase in misclassification error while both correct specification and overestimation of $d$ yield comparable classification accuracy to DSDA. Moreover, overestimating $d$ maintains feature selection sizes closer to the true sparsity level, providing a more parsimonious model. Based on these findings, we recommend using the eigenvalue-ratio rule described in Section 4 as a practical, reproducible approach to selecting $d$ while yielding a feature selection size much closer to the true model.
>
> For (3): {\bf Sparsity level $s$:} The sparsity parameter $s$ is selected via five-fold cross-validation over the grid $\{1,2,\dots,30\}$. The choice of $30$ is motivated by the sparsity condition $s = o((n/\log p)^\tau)$, for example, $\tau = 1/2$ as in Tony Cai's (2011) work where $n/\log p \approx 30$ in our setting. \textbf{Threshold $\eta_{\mathrm{thresh}}$:} Following  Jiang et al. (2018), $\eta_{\mathrm{thresh}}$ is chosen to minimize the in-sample classification error. Under the spiked covariance model, the grid search is restricted to a neighborhood around $$-2\log\left(\frac{|\widehat{\mathcal{W}}_2|}{|\widehat{\mathcal{W}}_1|}\right) - 2\log\left(\frac{n_1}{n_2}\right)$$, where the log-determinant ratio exhibits increased stability due to the consistent estimation of the bulk eigenvalues and accurate recovery of the leading spikes. We have added clarifying text in the main manuscript to emphasize these points and improve reproducibility in Section 4 and at the end of the $\mathrm{qda} \circ \mathrm{pca}$ Algorithm.
>
> For (4): Thank you for your comment. We have added a detailed comparison of computational complexity in Section 2.4.2. As discussed there, the proposed method is particularly efficient in the presence of low-rank or latent variable structures, whereas sparse $\beta$ baselines can incur higher computational cost in such settings due to iterative $\ell_{1}$-penalized optimization.

---

> > ### Comment · Reviewer_fxez · 2025-11-24
> > **Response to Authors Paper5423**
> >
> > I would like to thank the authors for their detailed response. I feel like my questions have been answered and my comments have been thoughtfully addressed. I appreciate the edits made in the revised version.

---

### Comment · Action_Editor_Tx4o · 2025-10-09
**Note regarding reviews**

Dear Authors,

One of the reviewers (fxez) accidentally left their review as an "official comment" on OpenReview as opposed to a review. So please respond to that official comment as if it were their review.

Thank you!

---

### Author Response · Authors · 2025-10-13

We thank all reviewers for their thoughtful and constructive feedback. We have carefully addressed each comment and revised the manuscript accordingly. Below, we provide detailed, point-by-point responses.

---

### Decision · Action_Editor_Tx4o · 2025-11-24

**Recommendation:** Accept as is

**Additional Comments:**

No changes are necessary.

**Audience:**

Yes

**Audience Explanation:**

The reviewers all agree that the results of the paper would be interesting to some of TMLR's audience. All reviewers agree that despite its technicality, the paper and its results are written clearly.

**Claims And Evidence:**

Yes

**Claims Explanation:**

The manuscript makes very clear claims regarding the misclassification error of lda composed with pca, and justifies them with primarily theoretical evidence (supported by corroborating empirical evidence). The paper demonstrates a comprehensive understanding of where this work is situated in the literature.